# RELIABILITY OF CKA AS A SIMILARITY MEASURE IN DEEP LEARNING

**MohammadReza Davari** [1,3] *   **Stefan Horoi** [2,3] *   **Amine Natik** [2,3]
**Guillaume Lajoie** [2,3]   **Guy Wolf** [2,3] †   **Eugene Belilovsky** [1,3] †

[1] Concordia University   [2] Université de Montréal   [3] Mila – Quebec AI Institute
{mohammadreza.davari, eugene.belilovsky}@concordia.ca
{stefan.horoi, amine.natik, guillaume.lajoie, guy.wolf}@umontreal.ca

## ABSTRACT

Comparing learned neural representations in neural networks is a challenging but important problem, which has been approached in different ways. The Centered Kernel Alignment (CKA) similarity metric, particularly its linear variant, has recently become a popular approach and has been widely used to compare representations of a network's different layers, of architecturally similar networks trained differently, or of models with different architectures trained on the same data. A wide variety of conclusions about similarity and dissimilarity of these various representations have been made using CKA. In this work we present analysis that formally characterizes CKA sensitivity to a large class of simple transformations, which can naturally occur in the context of modern machine learning. This provides a concrete explanation of CKA sensitivity to outliers, which has been observed in past works, and to transformations that preserve the linear separability of the data, an important generalization attribute. We empirically investigate several sensitivities of the CKA similarity metric, demonstrating situations in which it gives unexpected or counter-intuitive results. Finally we study approaches for modifying representations to maintain functional behaviour while changing the CKA value. Our results illustrate that, in many cases, the CKA value can be easily manipulated without substantial changes to the functional behaviour of the models, and call for caution when leveraging activation alignment metrics.

## 1 INTRODUCTION

In the last decade, increasingly complex deep learning models have dominated machine learning and have helped us solve, with remarkable accuracy, a multitude of tasks across a wide array of domains. Due to the size and flexibility of these models it has been challenging to study and understand exactly *how* they solve the tasks we use them on. A helpful framework for thinking about these models is that of *representation learning*, where we view artificial neural networks (ANNs) as learning increasingly complex internal representations as we go deeper through their layers. In practice, it is often of interest to analyze and compare the representations of multiple ANNs. However, the typical high dimensionality of ANN internal representation spaces makes this a fundamentally difficult task.

To address this problem, the machine learning community has tried finding meaningful ways to compare ANN internal representations and various *representation (dis)similarity measures* have been proposed (Li et al., 2015; Wang et al., 2018; Raghu et al., 2017; Morcos et al., 2018). Recently, Centered Kernel Alignment (CKA) (Kornblith et al., 2019) was proposed and shown to be able to reliably identify correspondences between representations in architecturally similar networks trained on the same dataset but from different initializations, unlike past methods such as linear regression or CCA based methods (Raghu et al., 2017; Morcos et al., 2018). While CKA can capture different notions of similarity between points in representation space by using different kernel functions, it was empirically shown in the original work that there are no real benefits to using CKA with a nonlinear kernel over its linear counterpart. As a result, linear CKA has been the preferred representation similarity measure of the machine learning community in recent years and other similarity measures (including nonlinear CKA) are seldomly used. CKA has been utilized in a number of works to make conclusions regarding the similarity between different models and their behaviours such as wide versus deep ANNs (Nguyen et al., 2021) and transformer versus CNN based ANNs (Raghu et al., 2021). They have also been used to draw conclusions about transfer learning (Neyshabur et al., 2020) and catastrophic forgetting (Ramasesh et al., 2021). Due to this widespread use, it is important to

---

*Equal contribution, name order randomized.†Equal senior-author contribution

understand how reliable the CKA similarity measure is and in what cases it fails to provide meaningful results. In this paper, we study CKA sensitivity to a class of simple transformations and show how CKA similarity values can be directly manipulated without noticeable changes in the model final output behaviour. In particular our contributions are as follows:

In Sec. 3 and with Thm. 1 we characterize CKA sensitivity to a large class of simple transformations, which can naturally occur in ANNs. With Cor. 3 and 4 we extend our theoretical results to cover CKA sensitivity to outliers, which has been empirically observed in previous work (Nguyen et al., 2021; Ding et al., 2021; Nguyen et al., 2022), and to transformations preserving linear separability of data, an important characteristic for generalization. Concretely, our theoretical contributions show how the CKA value between two copies of the same set of representations can be significantly decreased through simple, functionality preserving transformations of one of the two copies. In Sec. 4 we empirically analyze CKA's reliability, illustrating our theoretical results and subsequently presenting a general optimization procedure that allows the CKA value to be heavily manipulated to be either high or low without significant changes to the functional behaviour of the underlying ANNs. We use this to revisit previous findings (Nguyen et al., 2021; Kornblith et al., 2019).

## 2 BACKGROUND ON CKA AND RELATED WORK

**Comparing representations** Let $X \in \mathbb{R}^{n \times d_1}$ denote a set of ANN internal representations, i.e., the neural activations of a specific layer with $d_1$ neurons in a network, in response to $n \in \mathbb{N}$ input examples. Let $Y \in \mathbb{R}^{n \times d_2}$ be another set of such representations generated by the same input examples but possibly at a different layer of the same, or different, deep learning model. It is standard practice to center these representations column-wise (feature or "neuron" wise) before analyzing them. We are interested in representation similarity measures, which try to capture a certain notion of similarity between $X$ and $Y$.

**Quantifying similarity** Li et al. (2015) have considered one-to-one, many-to-one and many-to-many mappings between neurons from different neural networks, found through activation correlation maximization. Wang et al. (2018) extended that work by providing a rigorous theory of neuron activation subspace match and algorithms to compute such matches between neurons. Alternatively, Raghu et al. (2017) introduced SVCCA where singular value decomposition is used to identify the most important directions in activation space. Canonical correlation analysis (CCA) is then applied to find maximally correlated singular vectors from the two sets of representations and the mean of the correlation coefficients is used as a similarity measure. In order to give less importance to directions corresponding to noise, Morcos et al. (2018) introduced projection weighted CCA (PWCCA). The PWCCA similarity measure corresponds to the weighted sum of the correlation coefficients, assigning more importance to directions in representation space contributing more to the output of the layer. Many other representation similarity measures have been proposed based on linear classifying probes (Alain & Bengio, 2016; Davari et al., 2022), fixed points topology of internal dynamics in recurrent neural networks (Sussillo & Barak, 2013; Maheswaranathan et al., 2019), solving the orthogonal Procrustes problem between sets of representations (Ding et al., 2021; Williams et al., 2021) and many more (Laakso & Cottrell, 2000; Lenc & Vedaldi, 2018; Arora et al., 2017). We also note that a large body of neuroscience research has focused on comparing neural activation patterns in biological neural networks (Edelman, 1998; Kriegeskorte et al., 2008; Williams et al., 2021; Low et al., 2021).

**CKA** Centered Kernel Alignment (CKA) (Kornblith et al., 2019) is another such similarity measure based on the Hilbert-Schmidt Independence Criterion (HSIC) (Gretton et al., 2005) that was presented as a means to evaluate independence between random variables in a non-parametric way. For $K_{i,j} = k(x_i, x_j)$ and $L_{i,j} = l(y_i, y_j)$ where $k, l$ are kernels and for $H = I - \frac{1}{n}\mathbf{1}\mathbf{1}^\top$ the centering matrix, HSIC can be written as: $\text{HSIC}(K, L) = \frac{1}{(n-1)^2} tr(KHLH)$. CKA can then be computed as:

$$\text{CKA}(K, L) = \frac{\text{HSIC}(K, L)}{\sqrt{\text{HSIC}(K, K)\text{HSIC}(L, L)}} \tag{1}$$

In the linear case $k$ and $l$ are both the inner product so $K = XX^\top$, $L = YY^\top$ and we use the notation $CKA(X, Y) = CKA(XX^\top, YY^\top)$. Intuitively, HSIC computes the similarity structures of $X$ and $Y$, as measured by the kernel matrices $K$ and $L$, and then compares these similarity structures (after centering) by computing their alignment through the trace of $KHLH$.

**Recent CKA results** CKA has been used in recent years to make many claims about neural network representations. Nguyen et al. (2021) used CKA to establish that parameter initialization drastically impact feature similarity and that the last layers of overparameterized (very wide or deep) models

learn representations that are very similar, characterized by a visible "block structure" in the networks CKA heatmap. CKA has also been used to compare vision transformers with convolutional neural networks and to find striking differences between the representations learned by the two architectures, such as vision transformers having more uniform representations across all layers (Raghu et al., 2021). Ramasesh et al. (2021) have used CKA to find that deeper layers are especially responsible for forgetting in transfer learning settings.

Most closely related to our work, Ding et al. (2021) demonstrated that CKA lacks *sensitivity* to the removal of low variance principal components from the analyzed representations even when this removal significantly decreases probing accuracy. Also, Nguyen et al. (2022) found that the previously observed high CKA similarity between representations of later layers in large capacity models (so-called block structure) is actually caused by a few dominant data points that share similar characteristics. Williams et al. (2021) discussed how CKA does not respect the triangle inequality, which makes it problematic to use CKA values as a similarity measure in downstream analysis tasks. We distinguish ourselves from these papers by providing theoretical justifications to CKA sensitivity to outliers and to directions of high variance which were only empirically observed in Ding et al. (2021); Nguyen et al. (2021). Secondly, we do not only present situations in which CKA gives unexpected results but we also show *how* CKA values can be manipulated to take on arbitrary values.

**Nonlinear CKA** The original CKA paper (Kornblith et al., 2019) stated that, in practice, CKA with a nonlinear kernel gave similar results as linear CKA across the considered experiments. Potentially as a result of this, all subsequent papers which used CKA as a neural representation similarity measure have used linear CKA (Maheswaranathan et al., 2019; Neyshabur et al., 2020; Nguyen et al., 2021; Raghu et al., 2021; Ramasesh et al., 2021; Ding et al., 2021; Williams et al., 2021; Kornblith et al., 2021), and to our knowledge, no published work besides Kornblith et al. (2019); Nguyen et al. (2022) has used CKA with a nonlinear kernel. Consequently, we largely focus our analysis on linear CKA which is the most popular method and the one actually used in practice. However, our empirical results suggest that many of the observed problems hold for CKA with an RBF kernel and we discuss a possible way of extending our theoretical results to the nonlinear case.

## 3 CKA SENSITIVITY TO SUBSET TRANSLATION

Invariances and sensitivities are important theoretical characteristics of representation similarity measures since they indicate what kind of transformations respectively maintain or change the value of the similarity measure. In other words they indicate what transformations "preserve" or "destroy" the similarity between representations as defined by the specific measure used. Kornblith et al. (2019) argued that representation similarity measures should not be invariant to invertible linear transformations. They showed that measures which are invariant to such transformations give the same result for any set of representations of width (i.e. number of dimensions/neurons) greater than or equal to the dataset size. They instead introduced the CKA method, which satisfies weaker conditions of invariance, specifically it is invariant to orthogonal transformations such as permutations, rotations and reflections and to isotropic scaling. Alternatively, transformations to which representation similarity measures are sensitive have not been studied despite being highly informative as to what notion of "similarity" is captured by a given measure. For example, a measure that is highly sensitive to a transformation $\mathcal{T}$ is clearly measuring a notion of similarity that is destroyed by $\mathcal{T}$. In this section, we theoretically characterize CKA sensitivity to a wide class of simple transformations, namely the translation of a subset of the representations. We also justify why this class of transformations and the special cases it contains are important in the context of predictive tasks that are solved using neural networks.

**Theorem 1.** *Consider a set of $n$ internal representations in $p$ dimensions $X \in \mathbb{R}^{n \times p}$ that have been centered column-wise, let $S \subset X$ such that $\rho = \frac{|S|}{|X|} \leq \frac{1}{2}$ and $\vec{v}$ such that $\|\vec{v}\| = 1$. We define $X_{S,\vec{v},c} = S \cup \{x + c\vec{v} : x \in X \backslash S\}$. Then we have:*

$$\lim_{c \to \infty} CKA_{lin}(X, X_{S,\vec{v},c}) = \Gamma(\rho) \frac{\|\mathbb{E}_{x \in S}[x]\|^2}{\mathbb{E}_{x \in X}[\|x\|^2]} \sqrt{\dim_{PR}(X)} \tag{2}$$

*where $\Gamma(\rho) = \frac{\rho}{1-\rho} \in (0,1]$, and $\dim_{PR}(X) \triangleq \frac{(\sum_i \lambda_i)^2}{\sum_i \lambda_i^2} \in [1,p]$ the dimensionality estimate provided by the participation ratio of eigenvalues $\{\lambda_i\}$ of the covariance of $X$.*

**Corollary 2.** *Thm. 1 holds even if $S$ is taken such that $\rho = \frac{|S|}{|X|} \in (0.5, 1)$.*

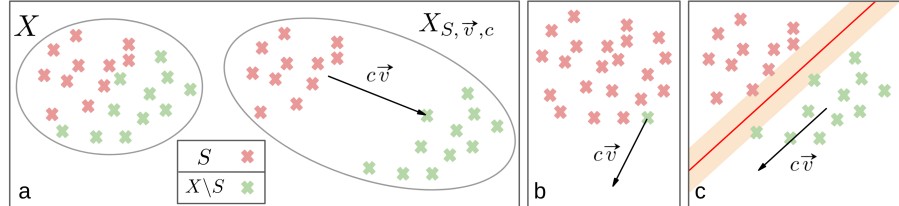

Figure 1: Visual representations of the transformations considered in the theoretical results. **a) Thm. 1:** The original set of neural representations $X$ contains subsets $S$ (red) and $X\backslash S$ (green). We can then build $X_{S,\vec{v},c}$ as a copy of $X$, where the points in $X\backslash S$ are translated a distance $c$ in direction $\vec{v}$. The linear CKA value between $X$ and $X_{S,\vec{v},c}$ is then computed. **b) Cor. 3:** $X$ and $X_{S,\vec{v},c}$ differ by a single point, which has been translated by $c\vec{v}$ in $X_{S,\vec{v},c}$. **c) Cor. 4:** $S$ and $X\backslash S$ are linearly separable (red line with orange margins), the transformation made to obtain $X_{S,\vec{v},c}$ preserves the linear separability of the data as well as the margins.

Our main theoretical result is presented in Thm. 1 and Cor. 2, whose proofs are provided in the Appendix (see Appendix C) along with additional details and further analysis (see Appendix D). These show that any set of internal neural representations $X$ (e.g., from hidden layers of a network) can be manipulated with simple transformations (translations of a subset, see Fig. 1.a) to significantly reduce the CKA between the original and manipulated set. We note that our theoretical results are entirely class and direction agnostic (except for Cor. 4 which isn't direction agnostic), see the last paragraph of Sec. 4.2 for more details on this point.

Specifically, we consider a copy of $X$ to which we apply a transformation where the representations of a subset $X\backslash S$ of the data is moved a distance $c$ along direction $\vec{v}$, resulting in the modified representation set $X_{S,\vec{v},c}$. A closed form solution is found for the limit of the linear CKA value between $X$ and $X_{S,\vec{v},c}$ as $c$ tends to infinity. We note that up to orthogonal transformations (which CKA is invariant to) the transformation $X \mapsto X_{S,\vec{v},c}$ is not difficult to implement when transforming representations between hidden layers in neural networks. More importantly, it is also easy to eliminate or ignore in a single layer transformation, as long as the weight vectors associated with the neurons in the subsequent layer are orthogonal to $\vec{v}$. Therefore, our results show that from a theoretical perspective, CKA can easily provide misleading information by capturing representation differences introduced by a shift of the form $X \mapsto X_{S,\vec{v},c}$ (especially with high magnitude $c$), which would have no impact on network operations or their effective task-oriented data processing.

The terms in Eq. 2 can each be analyzed individually. $\Gamma(\rho)$ depends entirely on $\rho$, the proportion of points in $X_{S,\vec{v},c}$ that have not been translated i.e. that are exactly at the same place as in $X$. Its value is between 0 and 1 and it tends towards 0 for small sizes of $S$. The participation ratio, with values in $[1,p]$, is used as an effective dimensionality estimate for internal representations (Mingzhou Ding & Dennis Glanzman, 2011; Mazzucato et al., 2016; Litwin-Kumar et al., 2017). It has long been observed that the effective dimensionality of internal representations in neural networks is far smaller than the actual number of dimensions of the representation space (Farrell et al., 2019; Horoi et al., 2020). $\mathbb{E}_{x \in X}[\|x\|^2]$ and $\|\mathbb{E}_{x \in S}[x]\|^2$ are respectively the average squared norms of all representations in $X$ and the squared norm of the mean of $S$, the subset of representations that are not being translated. Since most neural networks are trained using weight decay, the network parameters, and hence the resulting representations as well as these two quantities are biased towards small values in practice.

**CKA sensitivity to outliers** As mentioned in Sec. 2, it was recently found that the block structure in CKA heatmaps of high capacity models was caused by a few dominant data points that share similar characteristics (Nguyen et al., 2022). Other work has empirically highlighted CKA's sensitivity to directions of high variance, failing to detect important, function altering changes that occur in all but the top principal components (Ding et al., 2021). Cor. 3 provides a concrete explanation to these phenomena by treating the special case of Thm. 1 where only a single point, $\hat{x}$ is moved and thus has a different position in $X_{S,\vec{v},c}$ with respect to $X$, see Fig. 1.b for an illustration. We note that the term "subset translation" was coined by us and wasn't used in past work. However, all the papers referenced in this paragraph and later in this section present naturally occurring examples of subset translations in a set of representations relative to another, comparable set.

**Corollary 3.** *Thm. 1 holds in the special case where $S = \{\hat{x}\}$ is a single point, i.e. an outlier.*

Cor. 3 exposes a key sensitivity of linear CKA, namely its sensitivity to outliers. Consider two sets of representations that are identical in all aspects except for the fact that one of them contains an outlier, i.e. a representation further away from the others. Cor. 3 then states that as the difference

between the outlier's position in the two sets of representations becomes large the CKA value between the two sets drops dramatically, indicating high dissimilarity. Indeed, as previously noted, $\frac{\|\mathbb{E}_{x \in S}[x]\|^2}{\mathbb{E}_{x \in X}[\|x\|^2]} \sqrt{\dim_{PR}(X)}$ will be of relatively small value in practice so the whole expression in Eq. 2 will be dominated by $\Gamma(\rho) = \frac{\rho}{1-\rho}$. In the outlier case $\Gamma(\rho) \approx \rho = \frac{1}{\# \text{ representations}}$ which will be extremely small since for most modern deep learning datasets the number of examples in both the training and test sets is in the tens of thousands or more. This will drastically lower the CKA value between the two considered representations despite their obvious similarity.

**CKA sensitivity to transformations preserving linear separability**   Classical machine learning theory highlights the importance of data separability and of margin size for predictive models generalization (Lee et al., 1995; Bartlett & Shawe-Taylor, 1999). Large margins, i.e. regions surrounding the separating hyperplane containing no data points, are associated with less overfitting, better generalization and greater robustness to outliers and to noise. The same concepts naturally arise in the study of ANNs with past work establishing that internal representations become almost perfectly linearly separable by the network's last layer (Zeiler & Fergus, 2014a; Oyallon, 2017; Jacobsen et al., 2018; Belilovsky et al., 2019; Davari & Belilovsky, 2021). Furthermore, the quality of the separability, the margin size and the decision boundary smoothness have all been linked to generalization in neural networks (Verma et al., 2019). Given the theoretical and practical importance of these concepts and their natural prevalence in deep learning models it is reasonable to assume that a meaningful way in which two sets of representations can be "similar" is if they are linearly separable by the same hyperplanes in representation space and if their margins are equally as large. This would suggest that the exact same linear classifier could accurately classify both sets of representations. Cor. 4 treats this exact scenario as a special case of Thm. 1, see Fig. 1.c for an illustration. If $X$ contains two linear separable subsets, $S$ and $X \backslash S$, we can create $X_{S, \vec{v}, c}$ by translating one of the subsets in a direction that preserves the linear separability of the representations and the size of the margins while simultaneously decreasing the CKA between the original and the transformed representations, counterintuitively indicating a low similarity between representations.

**Corollary 4.** *Assume $S$ and $X \backslash S$ are linearly separably i.e. $\exists w \in \mathbb{R}^p$, the separating hyperplane's normal vector, and $k \in \mathbb{R}$ such that for every representation $x \in X$ we have: $x \in S \Rightarrow \langle w, x \rangle \leq k$ and $x \in X \backslash S \Rightarrow \langle w, x \rangle > k$. We can then pick $\vec{v}$ such that $S$ and $\{x + c\vec{v} : x \in X \backslash S\}$ are linearly separable by the exact same hyperplane and with the exact same margins as $S$ and $X \backslash S$ for any value of $c \in \mathbb{R}_{\geq 0}$ and Thm. 1 still holds.*

**Extensions to nonlinear CKA**   As previously noted in Sec. 2, given the popularity of linear CKA, it is outside the scope of our work to theoretically analyze nonlinear kernel CKA. However one can consider extending our theoretical results to the nonlinear CKA case with symmetric, positive definite kernels. Indeed we know from reproducing kernel Hilbert space (RKHS) theory that we can write such a kernel as an inner product in an implicit Hilbert space. While directly translating points in the representations space would likely not drive CKA values down as in the linear case, it would suffice to find/learn which transformations in representations space correspond to translations in the implicit Hilbert space. Our results should hold if we apply the found transformations, instead of translations, to a subset of the representations. Although practically harder to implement than simple translations, we hypothesize that it would be possible to learn such transformations.

## 4 EXPERIMENTS AND RESULTS

In this section we will demonstrate several counterintuitive results, which illustrate cases where similarity measured by CKA is inconsistent with expectations (Sec 4.1) or can be manipulated without noticeably changing the functional behaviour of a model (Sec. 4.2 and 4.3). We emphasize that only Sec. 4.2 is directly tied to the theoretical results presented in the Sec. 3. On the other hand, sections 4.1 and 4.3 discuss empirical results which are not explicitly related to the theoretical analysis.

### 4.1 CKA EARLY LAYER RESULTS

CKA values are often treated as a surrogate metric to measure the usefulness and similarity of a network's learned features when compared to another network (Ramasesh et al., 2021). In order to analyze this common assumption, we compare the features of: (1) a network trained to generalize on the CIFAR10 image classification task (Krizhevsky et al., 2009), (2) a network trained to "memorize" the CIFAR10 images (i.e. target labels are random), and (3) an untrained randomly initialized network (for network architecture and training details see the Appendix). As show in Fig. 2, early layers of these networks should have very similar representations given the high CKA values. Under the previously presented assumption, one should therefore conclude that the learned features at these

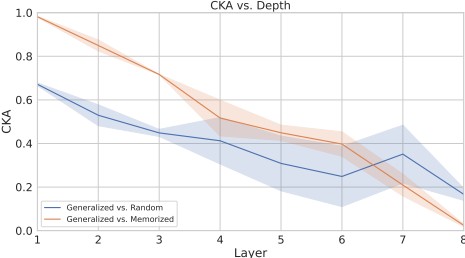

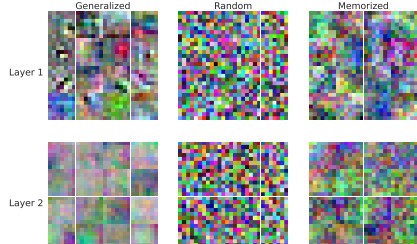

Figure 2: A layer-wise comparison based on the value of the CKA between a generalized, memorized, and randomly populated network. This comparison reveals that early layers of these networks achieve relatively high CKA values. Mean and standard deviation across 5 seeds are shown.

Figure 3: The convolution filters within the first two layers of a generalized, memorized, and a randomly initialized network. Each of 16 squares corresponds to a filter. Layer 1 filter colors correspond to RGB, Layer 2 for each filter we randomly samples 3 channels for visualization. This elucidates that the features are (1) drastically different, and (2) not equally useful despite the CKA results in Fig. 2

layers are relatively similar and equally valuable. However this is not the case, we can see in Fig. 3 that the convolution filters are drastically different across the three networks. Moreover, Fig. 3 elucidates that considerably high CKA similarity values for early layers, does not necessarily translate to more useful, or similar, captured features. In Fig. 16 of the Appendix we use linear classifying probes to quantify the "usefulness" of learned features and those results further support this claim.

## 4.2 PRACTICAL IMPLICATIONS OF THEORETICAL RESULTS

Here we empirically test the behaviour of linear and RBF CKA in situations inspired by our theoretical analysis, first in an artificial setting, then in a more realistic one. We begin with artificially generated representations $X \in \mathbb{R}^{n \times d}$ to which we apply subset translations to obtain $Y \in \mathbb{R}^{n \times d}$, similar to what is described in Thm. 1. We generate $X$ by sampling 10K points uniformly from the 1K-dimensional unit cube centered at the origin and 10K points from a similar cube centered at $(1.1, 0, 0, \ldots, 0)$, so the points from the two cubes are linearly separable along the first dimension. We translate the representations from the second cube in a random direction sampled from the $d$-dimensional ball and we plot the CKA values between $X$ and $Y$ as a function of the translation distance in Fig. 4a. This transformation entirely preserves the topological structure of the representations as well as their local geometry since the points sampled from each cube have not moved with respect to the other points sampled from the same cube and the two cubes are still separated, only the distance between them has been changed. Despite these multiple notions of "similarity" between $X$ and $Y$ being preserved, the CKA values quickly drop below 0.2 for both linear and RBF CKA. While our theoretical results (Thm. 1) predicted this drop for linear CKA, it seems that RBF CKA is also highly sensitive to translations of a subset of the representations. Furthermore, it is surprising to see that the drop in CKA value occurs even for relatively small translation distances. We note that RBF CKA with $\sigma$ equal to 0.2 times the median distance between examples is unperturbed by the considered transformation. However, as we observe ourselves (RBF CKA experiments in the supplement) and as was found in the original CKA paper (see Table 2 of Kornblith et al. (2019)), RBF CKA with $\sigma = 0.2 \times$ median is significantly less informative than RBF CKA with higher values of $\sigma$. With small values of $\sigma$, RBF CKA only captures very local, possibly trivial, relationships.

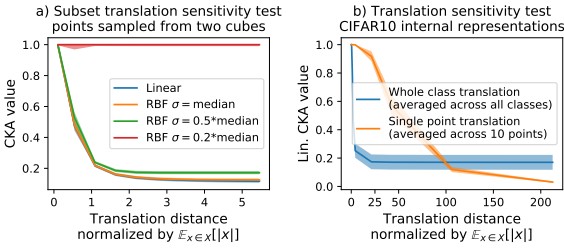

Figure 4: **a)** Linear and RBF CKA values between the artificial representations $X$ and the subset translated version $Y$ as a function of the translation distance. **b)** CKA value between a CNN's internal representations of the CIFAR10 training set and modified versions where either a class or a single point is translated as functions of the translation distance. The translation distances are normalized by the average norm of the unmodified representations, $\mathbb{E}_{x \in X}[\|x\|]$.

In a more realistic setting we test the practical implications of linear CKA sensitivity to outliers (see Cor. 3) and to transformations that preserve the linear separability of the data as well as the margins (see Cor. 4). We consider the 9 layers CNN presented in Sec. 6.1 of Kornblith et al. (2019) trained on CIFAR10. As argued in Sec. 2, when trained on classification tasks, ANNs tend to learn increasingly complex representations of the input data that can be almost perfectly linearly separated into classes by the last layer of the network. Therefore a meaningful way in which two sets of representations can

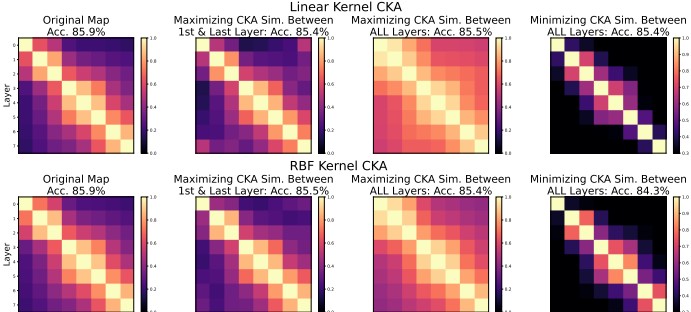

Figure 5: Original Map is the CKA map of a network trained on CIFAR10. We manipulate this network to produce CKA maps which: (1) maximizes the CKA similarity between the 1st and last layer, (2) maximizes the CKA similarity between all layers, and (3) minimizes the CKA similarity between all layers. In cases (1) and (2), the network experiences only a slight loss in performance, which counters previous findings by achieving a strong CKA similarity between early and late layers. We find similar results are easily achieved in the kernel CKA case.

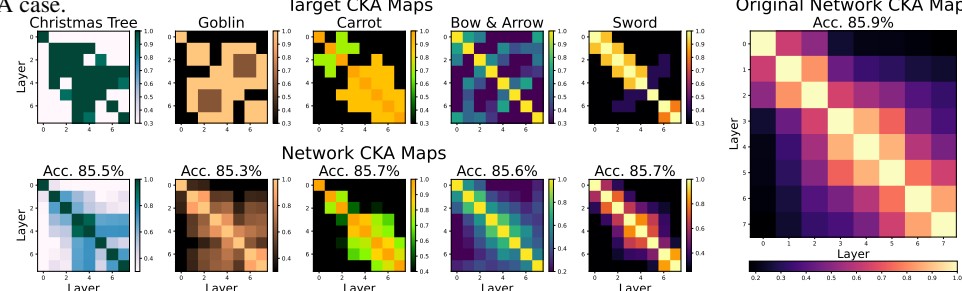

Figure 6: The comical target CKA maps (first row) are used as the objective for the CKA map loss in Eq. 3, while prioritizing network performance (small tolerance for changes in accuracy $\delta_{\mathrm{acc}}$). The second row shows the CKA map produced by the network.

be "similar" in practice is if they are linearly separable by the same hyperplanes in parameter space, with the same margins. Given $X$, the network's internal representations of 10k training images at the last layer before the output we can use an SVM classifier to extract the hyperplanes in parameter space which best separate the data (with approx. 91% success rate). We then create $Y$ by translating a subset of the representations in a direction which won't cross these hyperplanes, and won't affect the linear separability of the representations. We plot the CKA values between $X$ and $Y$ according to the translation distance in Fig. 4b. The CKA values quickly drop to 0, despite the existence of a linear classifier that can classify both sets of representations into the correct classes with $> 90\%$ accuracy. In Fig. 4b we also examine linear CKA's sensitivity to outliers. Plotted are the CKA values between the set of training image representations and the same representations but with a single point being translated from its original location. While the translation distance needed to achieve low CKA values is relatively high, the fact that the position of a *single* point out of *tens of thousands* can so drastically influence the CKA value raises doubts about CKA's reliability as a similarity metric.

We note that our main theoretical results, namely Th. 1, Cor. 2 and Cor. 3 are entirely class and direction agnostic. The empirical results presented in this section are simply examples that we deemed particularly important in the context of ML but the same results would hold with any subset of the representations and translation direction, even randomly chosen ones. This is important since the application of CKA is not restricted to cases where labels are available, for example it can also be used in unsupervised learning settings (Grigg et al., 2021). Furthermore, the subset translations presented here were added manually to be able to run the experiments in a controlled fashion but these transformations can naturally occur in ANNs, as discussed in Sec. 3, and one would not necessarily know that they have occurred. We also run experiments to evaluate CKA sensitivity to invertible linear transformations, see the Appendix for justification and results.

## 4.3 EXPLICITLY OPTIMIZING THE CKA MAP

The CKA map, commonly used to analyze network architectures (Kornblith et al., 2019) and their inter-layers similarities, is a matrix $M$, where $M[i, j]$ is the CKA value between the activations of layers $i$, and $j$ of a network. In many works (Nguyen et al., 2021; Raghu et al., 2021; Nguyen et al.,

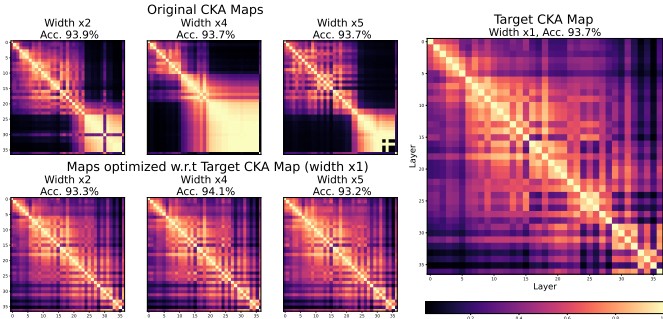

Figure 7: ResNet-34 networks of different widths and their corresponding CKA Maps are modified to produce CKA maps of thin networks. Top row **Original** shows the unaltered CKA map of the networks derived from "normal" training. **Optimized** shows the CKA map of the networks after their map is optimized to mimic the thin net *target* CKA map. See Appendix for more networks.

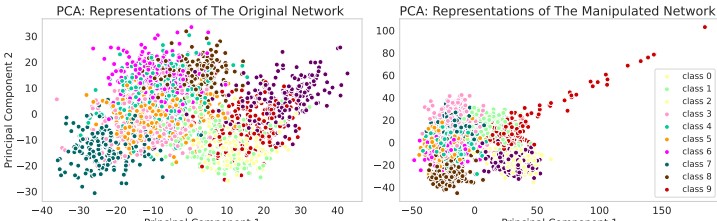

Figure 8: PCA of the networks presented in Fig. 5 before (left) and after (right) being optimized to manipulate the CKA map with Eq 3. Noticeably to achieve the objective the optimization displaces a subset of a single class.

2022) these maps are used explicitly to obtain insights on how different models behave, compared to one another. However, as seen in our analysis so far it is possible to manipulate the CKA similarity value, decreasing and increasing it without changing the behaviour of the model on a target task. In this section we set to directly manipulate the CKA map of a trained network $f_{\theta^*}$, by adding the desired CKA map, $M_{target}$, to its optimization objective, while maintaining its original outputs via distillation loss (Hinton et al., 2015). The goal is to determine if the CKA map can be changed while keeping the model performance the same, suggesting the behaviour of the network can be maintained while changing the CKA measurements. To accomplish this we optimize $f_\theta$ over the training set $(X, Y)$ (note however that the results are shown on the test set) via the following objective:

$$\theta_{new}^* = \operatorname{argmin}_\theta \left( \mathcal{L}_{distill}\left(f_{\theta^*}(X), f_\theta(X)\right) + \lambda \mathcal{L}_{map}\left(M_{f_\theta(X)}, M_{target}\right)\right) \tag{3}$$

where $\mathcal{L}_{map}\left(M_{f_\theta(X)}, M_{target}\right) = \sum_{i,j} \ln \cosh\left(M[i,j]_{f_\theta(X)} - M[i,j]_{target}\right)$. The $\lambda$ multiplier in Eq 3 is the weight that balances the two losses. Making $\lambda$ large will favour the agreements between the target and network CKA maps over preservation of the network outputs. In our experiments $\lambda$ is allowed to change dynamically at every optimization step. Using the validation set accuracy as a surrogate metric for how well the network's representations are preserved, $\lambda$ is then modulated to learn maps. If the difference between the original accuracy of the network and the current validation accuracy is above a certain threshold ($\delta_{\text{acc}}$) we scale down $\lambda$ to emphasize the alignment of the network output with the outputs of $f_{\theta^*}$, otherwise we scale it up to encourage finer agreement between the target and network CKA maps (see Appendix for the pseudo code).

Fig. 5 shows the CKA map of $f_{\theta^*}$ along with the CKA map of three scenarios we investigated: (1) maximizing the CKA similarity between the 1ˢᵗ and last layer, (2) maximizing the CKA similarity between all layers, and (3) minimizing the CKA similarity between all layers (for network architecture and training details see Appendix). In cases (1) and (2), the network performance is barely hindered by the manipulations of its CKA map. This is surprising and contradictory to the previous findings (Kornblith et al., 2019; Raghu et al., 2021) as it suggests that it is possible to achieve a strong CKA similarity between early and later layers of a well-trained network without noticeably changing the model behaviour. Similarly we observe that for the RBF kernel based CKA (Kornblith et al., 2019) we can obtain manipulated results using the same procedure. The bandwidth $\sigma$ for the RBF kernel CKA is set to 0.8 of the median Euclidean distance between representations (Kornblith et al., 2019). In the Appendix we also show similar analysis on other $\sigma$ values. We further experiment with manipulating the CKA map of $f_{\theta^*}$ to produce a series of comical CKA maps (Fig. 6) while maintaining similar model accuracy. Although the network CKA maps seen in Fig. 6 closely resemble

their respective targets, it should be noted that we prioritized maintaining the network outputs, and ultimately its accuracy by choosing small $\delta_{acc}$. Higher thresholds of accuracy result in stronger agreements between the target and network CKA maps at the cost of performance.

**Wider networks**  Nguyen et al. (2021) and Nguyen et al. (2022) studied the behaviour of wider and deeper networks using CKA maps, obtaining a block structure, which was subsequently used to obtain insights. We revisit these results and investigate whether the CKA map corresponding to a wider network can be mapped to a thin network. Our results for the CIFAR10 dataset and ResNet-34 (He et al., 2016) are shown in Fig. 7 (for details on the architecture and training procedure see Appendix). We observe that the specific structures associated with wider network can be completely removed and the map can be nearly identical to the thinner model without changing the performance substantially. Results on OOD datasets and with vision transformers yield similar results (appendix Fig. 10, 14).

**Analysis of Modified Representations**  Our focus in Sec 4.3 has been to use optimization to achieve a desired target CKA manipulations without any explicit specification of how to perform this manipulation. We perform an analysis to obtain insights on how representations are changed by optimizing Eq 3 in Fig. 8. Here using the modified network from case (2) of Fig. 5 we compute the PCA of the *test set*'s last hidden representation (whose CKA compared to the first layer is increased). We observe that a a single class has a very noticeable set of points that are translated in a particular direction, away from the general set of classes. This mechanism of manipulating the CKA aligns with our theoretical analysis. We emphasize, that in this case it is a completely emergent behavior.

## 5    DISCUSSION AND CONCLUSION

Given the recent popularity of linear CKA as a similarity measure in deep learning, it is essential to better study its reliability and sensitivity. In this work we have presented a formal characterization of CKA's sensitivity to translations of a subset of the representations, a simple yet large class of transformations that is highly meaningful in the context of deep learning. This characterization has provided a theoretical explanation to phenomena observed in practice, namely CKA sensitivity to outliers and to directions of high variance. Our theoretical analysis also show how the CKA value between two sets of representations can diminish even if they share local structure and are linearly separable by the same hyperplanes, with the same margins. This meaningful way in which two sets of representations can be similar, as justified by classical machine learning theory and seminal deep learning results, is therefore not captured by linear CKA. We further empirically showed that CKA attributes low similarity to sets of representations that are directly linked by simple affine transformations that preserve important functional characteristics and it attributes high similarity to representations from unalike networks. Furthermore, we showed that we can manipulate CKA in networks to result in arbitrarily low/high values while preserving functional behaviour, which we use to revisit previous findings (Nguyen et al., 2021; Kornblith et al., 2019).

It is not our intention to cast doubts over linear CKA as a representation similarity measure in deep learning, or over recent work using this method. Indeed, some of the problematic transformations we identify are not necessarily encountered in many applications. However, given the popularity of this method and the exclusive way it has been applied to compare representations in recent years, we believe it is necessary to better understand its sensitivities and the ways in which it can be manipulated. Our results call for caution when leveraging linear CKA, as well as other representations similarity measures, and especially when the procedure used to produce the model is not known, consistent, or controlled. An example of such a scenario is the increasingly popular use of open-sourced pre-trained models. Such measures are trying to condense a large amount of rich geometrical information from many high-dimensional representations into a single scalar $\in [0, 1]$. Significant work is still required to understand these methods in the context of deep learning representation comparison, what notion of similarity each of them concretely measures, to what transformations each of them are sensitive or invariant and how this all relates to the functional behaviour of the models being analyzed.

In the meantime, as an alternative to using solely linear CKA, we deem it prudent to utilize multiple similarity measures when comparing neural representations and to try relating the results to the specific notion of "similarity" each measure is trying to quantify. Comparison methods that are straightforward to interpret or which are linked to well understood and simple theoretical properties such as solving the orthogonal Procrustes problem (Ding et al., 2021; Williams et al., 2021) or comparing the sparsity of neural activations (Kornblith et al., 2021) can be a powerful addition to any similarity analysis. Further, data visualizations methods can potentially help to better understand the structure of neural representations in certain scenarios (e.g. Nguyen et al. (2019); Gigante et al. (2019); Horoi et al. (2020); Recanatesi et al. (2021)).

ACKNOWLEDGEMENTS

This work was partially funded by OpenPhilanthropy [M.D., E.B.]; NSERC CGS D, FRQNT B1X & UdeM A scholarships [S.H.]; NSERC Discovery Grant RGPIN-2018-04821 & Samsung Research Support [G.L.]; NSERC Discovery Grant 03267 [G.W.]; and Canada CIFAR AI Chairs [G.L., G.W.]. This work is also supported by resources from Compute Canada and Calcul Quebec. The content is solely the responsibility of the authors and does not necessarily represent the views of the funding agencies.

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

# A    EXPERIMENTAL DETAILS

## A.1    MINIBATCH CKA

In our experiments (with the exception of subsection 4.2), in order to reduce memory consumption, we use the minibatch implementation of the CKA similarity Nguyen et al. (2021; 2022). More precisely, let $\psi$ be a kernel function (we experiment with linear and RBF kernel), and $X_b \in \mathbb{R}^{m \times n_1}$ and $Y_b \in \mathbb{R}^{m \times n_2}$ be the minibatches of $m$ samples from two network layers containing $n_1$ and $n_2$ neurons respectively. We estimate the value of CKA by averaging the Hilbert-Schmidt independence criterion (HSIC), over all minibatches $b \in B$ via:

$$\text{CKA}_{\text{minibatch}} = \frac{\frac{1}{|B|} \sum_{b \in B} \text{HSIC}_1(Q_b, Z_b)}{\sqrt{\frac{1}{|B|} \sum_{b \in B} \text{HSIC}_1(Q_b, Q_b)} \sqrt{\frac{1}{|B|} \sum_{b \in B} \text{HSIC}_1(Z_b, Z_b)}} \tag{4}$$

Where $Q_b = \psi(X_b, X_b)$, $Z_b = \psi(Y_b, Y_b)$, and $\text{HSIC}_1$ is the unbiased estimator of HSIC Song et al. (2012), hence the value of the CKA is independent of the batch size.

## A.2    NETWORK ARCHITECTURE

In Sec. 4.1 we use a 9 layer neural network; the first 8 of these layers are convolution layers and the last layer is a fully connected layer used for classification. We use ReLU (Nair & Hinton, 2010) throughout the network. The kernel size of every convolution layer is set to $(3, 3)$ except the first two convolution layers, which have $(7, 7)$ kernels. All convolution layers follow a padding of 0 and a stride of 1. Number of kernels in each layer of the network, from the lower layers onward follows: $[16, 16, 32, 32, 32, 64, 64]$. In this network, every convolution layer is followed by batch normalization (Ioffe & Szegedy, 2015). The network we used in Sec. 4.3 to obtain Figures 5 and 6 is similar to the network we just described, except the kernel size for all layers are set to $(3, 3)$. For the **Wider networks** experiments in Sec. 4.3 we use a ResNet-34 (He et al., 2016) network, where we scale up the channels of the network to increase the width of the network (see Fig. 7).

## A.3    TRAINING DETAILS

The models in Sec. 4.1, both the generalized and memorized network, were trained for 100 epochs using AdamW (Loshchilov & Hutter, 2017) optimizer with a learning rate (LR) of $1e{-}3$ and a weight decay of $5e{-}4$. The LR is follows cosine LR scheduler (Loshchilov & Hutter, 2016) with an initial LR stated earlier.

The training of the base model (original) model in Sec. 4.3 seen in Figures 5 and 6 follows the same training procedure as of the models from Sec. 4.1, except in this setting we train the model for 200 epochs, with an initial LR of 0.01. All other models in Sec. 4.3 seen in Figures 5 and 6 (with a target CKA map to optimize) are also trained with similar training hyperparameters to that of the base model, except the followings: (1) these models are only trained for 30 epochs. (2) the objective function includes a hyperparameter $\lambda$ (see Eq. 3), which we initially set to 500 for all models and is changed dynamically following the Algo. 1 during the training by 0.8 on each iteration. (3) The cosine LR scheduler includes a warm-up step of 500 optimization steps. (4) the LR is set to $1e{-}3$ (4) The distillation loss in the objective function depends on a temperature parameter, which we set to 0.2.

The training procedure for the **Wider networks** experiments in Sec. 4.3 is similar to the previous training procedures in this section (Figures 5 and 6). Except that the *Original* models are trained for only 100 epochs and the *Optimized w.r.t Target Maps* models are trained for 15 epochs.

## A.4    CKA MAP LOSS BALANCE

Algo. 1 shows the pseudo code of the dynamical scaling of the $\lambda$ loss balance parameter seen in Eq. 3. Using the validation set accuracy as a surrogate metric for how well the network's representations are preserved, $\lambda$ is then modulated to learn maps. If the difference between the original accuracy of the network and the current validation accuracy is above a certain threshold ($\delta_{\text{acc}}$) we scale down $\lambda$ to emphasize the alignment of the network output with the outputs of $f_{\theta^*}$, otherwise we scale it up to encourage finer agreement between the target and network CKA maps.

---

**Algorithm 1:** Dynamical balancing of Distillation and CKA map loss in Eq. 3

---

**Data:** $100 \geq$ Original Accuracy $> 0; 100 \geq$ Current Validation Accuracy $> 0;$
Accuracy Threshold $\geq 0; 1 \geq$ Scaling Factor $> 0;$ Initial Lambda $> 0$
**Result:** $\lambda$
$\text{acc}_0 \leftarrow$ Original Accuracy;
$\text{acc}_1 \leftarrow$ Current Validation Accuracy;
$\eta \leftarrow$ Accuracy Threshold;
$\alpha \leftarrow$ Scaling Factor;
$\lambda \leftarrow$ Initial Lambda;
$\delta_{\text{acc}} \leftarrow \text{acc}_0 - \text{acc}_1;$
**if** $\delta_{\text{acc}} > \eta$ **then**
  | $\lambda \leftarrow \lambda \times \alpha$
**else**
  | $\lambda \leftarrow \lambda/\alpha$
**end**

**Return** $\lambda$;

---

## B  ADDITIONAL RESULTS

### B.1  CKA SENSITIVITY TO INVERTIBLE LINEAR TRANSFORMATIONS

We also experiment, for linear CKA, with a type of transformation that isn't considered by our theoretical results but which we deemed interesting to analyze empirically, namely multiplications by invertible matrices. Consider a matrix $M \in \mathbb{R}^{d \times d}$ whose elements are sampled from a Gaussian with mean $\mu$ and standard deviation $\sigma$. We verify the invertibility of $M$ since it is not guaranteed and only keep invertible matrices. We show the CKA values between $X$ and the transformed $Y = XM$ in Fig. 9). Since this is an invertible linear transformation we would expect it to only modestly change the representations in $X$ and the CKA value to be only slightly lower than 1. However, we observe that even for small values of $\mu$ and $\sigma$, CKA drops to 0, which would indicate that the two sets of representations are dissimilar and not linked by a simple, invertible transformation.

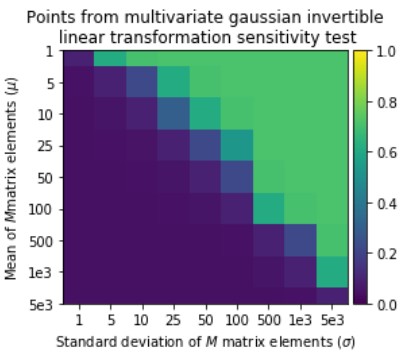

Figure 9: Linear CKA values between the artificial representations $X$ and $Y = XM$ with $M$ being an invertible matrix with elements sampled from $\mathcal{N}(\mu, \sigma^2)$ as a function of $\mu$ and $\sigma$. The mean and standard deviation across 10 random instantiations the translation direction and $M$ are shown.

While Th.1 of Kornblith et al. (2019) implies that invariance to invertible linear transformations is generally not a desirable property for ANN representation similarity measures, there are relatively common scenarios in which the hypotheses of the theorem are not necessarily respected, i.e. where the dataset size is larger than the width of the layer. Such is the case in smaller ANNs or even at the last layers of large models which are often fully connected and of far smaller size than the input space or the intermediate layers. Given these situations we see no reason to completely dismiss this invariance as being possibly desirable in certain, albeit not all, contexts.

### B.2  WIDER NETWORKS

In Sec. 4.3 we investigated whether the CKA of the wide networks can be mapped to a thin network (see Fig. 7) using ResNet-34 models and the CIFAR10 dataset. In Fig. 10, we use the same networks (trained on CIFAR10) and measure their CKA similarity maps under the Patch Camelyon dataset (Veeling et al., 2018). Patch Camelyon dataset contains histopathologic scans of lymph node sections, which is drastically different from the CIFAR10 dataset both in terms of pixel distribution and the semantics of the data. As we can see in Fig. 10, even under this drastic shift in data distribution

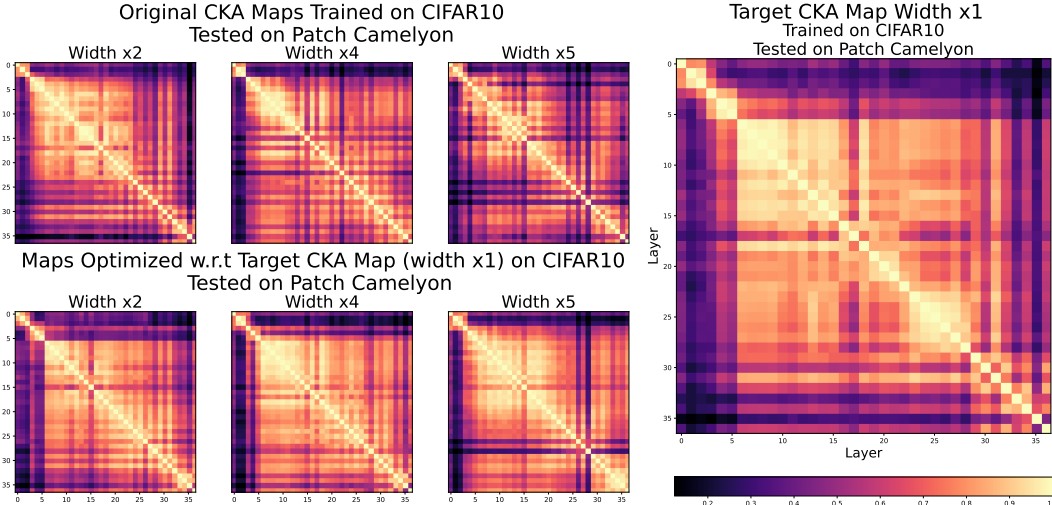

Figure 10: In Fig. 7 we are presented a series of ResNet-34 networks of different widths and their corresponding CKA Maps, which are modified to produce CKA maps of thin networks using the CIFAR10 dataset. We used these networks and measured their CKA maps using Patch Camelyon dataset. Top row **Original** shows the unaltered CKA map of the networks derived from "normal" training on CIFAR10, tested on Patch Camelyon dataset. **Optimized** shows the CKA map of the networks after their map is optimized to mimic the thin network *target* CKA map using CIFAR10, tested on Patch Camelyon dataset.

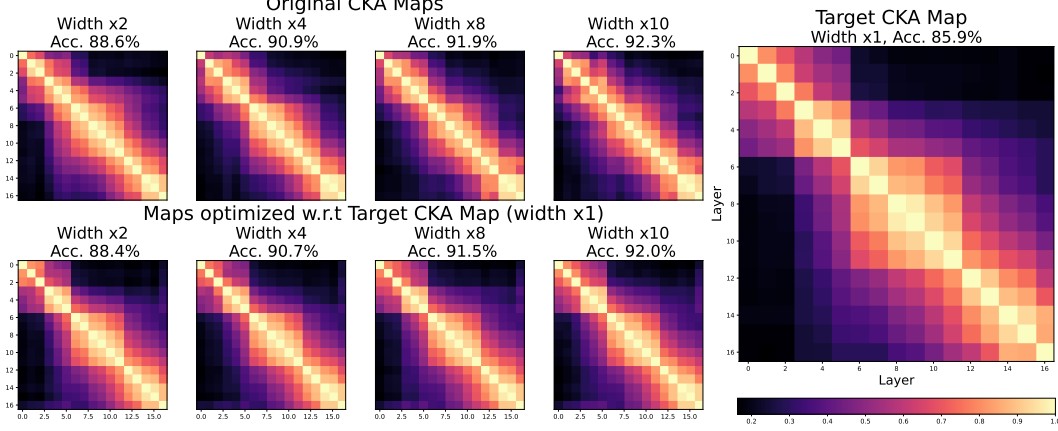

Figure 11: VGG style networks of different widths and their corresponding CKA Maps are modified to produce CKA maps of thin networks. Top row **Original** shows the unaltered CKA map of the networks derived from "normal" training. **Optimized** shows the CKA map of the networks after their map is optimized to mimic the thin net *target* CKA map.

the CKA maps of the networks *Optimized w.r.t Target CKA Map* resemble the CKA map of the thin target network, suggesting the generalizability of the CKA map optimization.

The network architecture presented in the **Wider networks** experiments seen in Sec. 4.3 is ResNet-34. We experimented with a VGG style network architecture to broaden our findings to other network architectures (see Sec. A.2 for details). As we can see in Fig. 11 we observe similar results to the ones shown in Fig. 7.

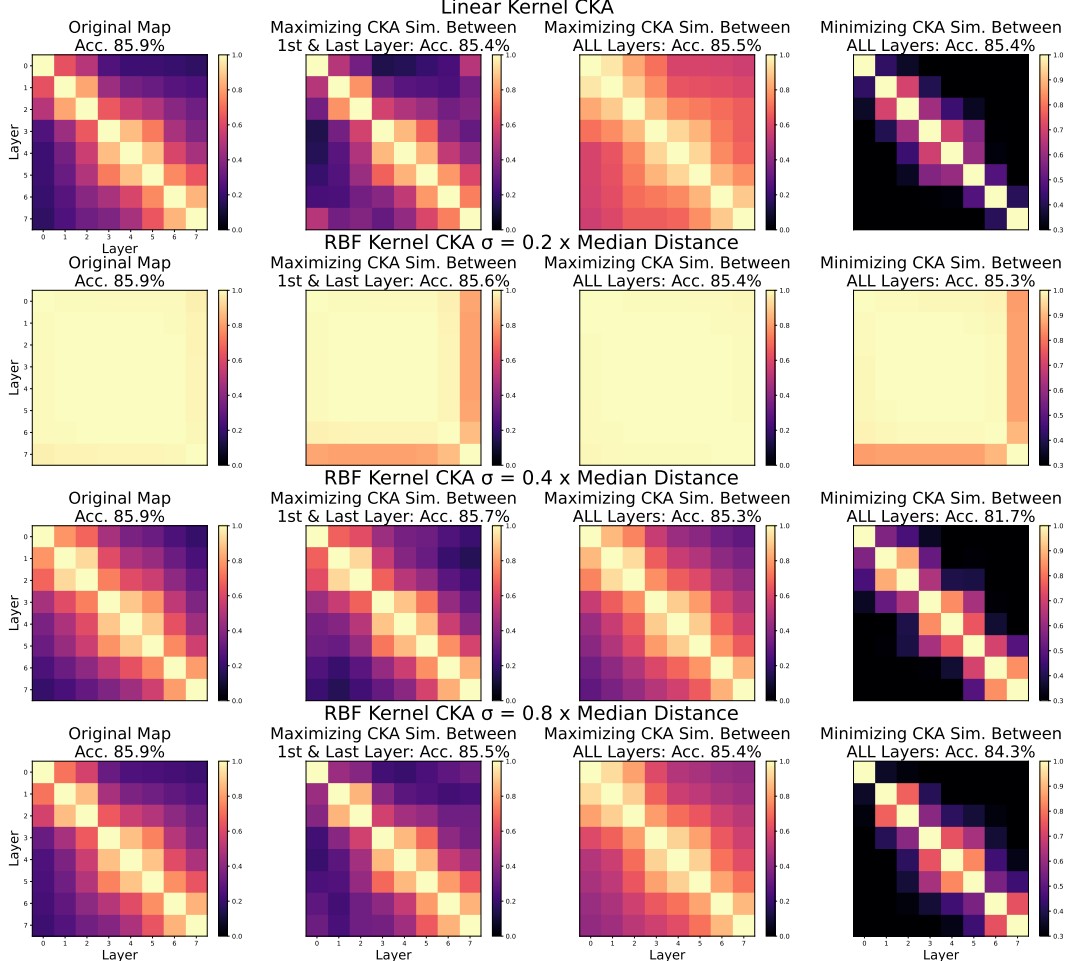

Figure 12: Original Map is the CKA map of a network trained on CIFAR10. We manipulate this network to produce CKA maps which: (1) maximizes the CKA similarity between the 1$^{st}$ and last layer, (2) maximizes the CKA similarity between all layers, and (3) minimizes the CKA similarity between all layers. In cases (1) and (2), the network experiences only a slight loss in performance, which counters previous findings by achieving a strong CKA similarity between early and late layers.

### B.3 RBF KERNEL

In Fig. 12 we extend our results shown in Fig. 5 to other bandwidth values commonly used for the RBF kernel CKA (Kornblith et al., 2019). When the CKA values are meaningful, we observe that the RBF kernel CKA values can be manipulated via the procedure described in Sec. 4.3.

### B.4 CKA MAP OPTIMIZATION VIA LOGISTIC LOSS

In Sec. 4.3, we manipulated a network's CKA map, while closely maintaining its outputs via the distillation loss seen in Eq. 3. However, a logistic loss also works in this setting, i.e. the substitution of the distillation loss with cross-entropy loss in Eq. 3 yields similar results. In Fig. 13, we repeated the linear CKA experiments seen in the first row of the Fig. 5 using cross-entropy loss instead of distillation loss.

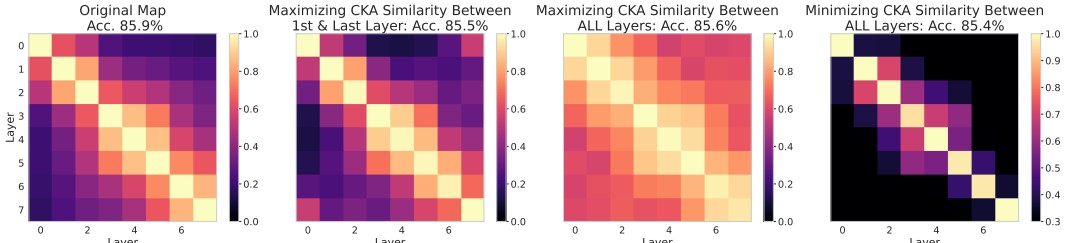

Figure 13: Original Map is the CKA map of a network trained on CIFAR10. We manipulate this network following a modified version of Eq. 3 (distillation loss is substituted with cross-entropy loss) to produce CKA maps which: (1) maximizes the CKA similarity between the 1$^{st}$ and last layer, (2) maximizes the CKA similarity between all layers, and (3) minimizes the CKA similarity between all layers. In cases (1) and (2), the network experiences only a slight loss in performance, which counters previous findings by achieving a strong CKA similarity between early and late layers.

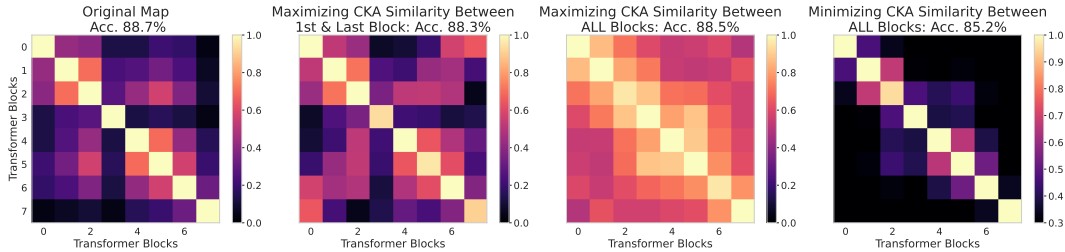

Figure 14: Original Map is the CKA map of a ViT (Dosovitskiy et al., 2020) network trained on CIFAR10. We manipulate this network following the Eq. 3 to produce CKA maps which: (1) maximizes the CKA similarity between the 1$^{st}$ and last Transformer block, (2) maximizes the CKA similarity between all Transformer blocks, and (3) minimizes the CKA similarity between all Transformer blocks.

## B.5  CKA OPTIMIZATION OF VIT

In Fig. 5, we manipulated the CKA map of a VGG style model trained on CIFAR10 in order to: (1) maximize the CKA similarity between the 1$^{st}$ and last layer, (2) maximize the CKA similarity between all layers, and (3) minimize the CKA similarity between all layers.

We further explored this setting at the model architecture level. Given the recent popularity of the Transformer (Vaswani et al., 2017) architecture in a variety of domains such as NLP Devlin et al. (2018); Farahnak et al. (2021); Raffel et al. (2020); Davari et al. (2020), Computer Vision Dosovitskiy et al. (2020); Zhou et al. (2021); Liu et al. (2021), and Tabular Huang et al. (2020); Arik & Pfister (2021), we implemented a Vision Transformer (ViT) (Dosovitskiy et al., 2020) style model for the CIFAR10 dataset, containing 8 Transformer (Vaswani et al., 2017) blocks (see other architectural details in Tab. 1) in order to: (1) maximize the CKA similarity between the 1$^{st}$ and last Transformer block, (2) maximize the CKA similarity between all Transformer blocks, and (3) minimize the CKA similarity between all Transformer blocks. As we can see in Fig. 14 these manipulations are achieved with minimal loss of performance, which underlines the model-agnostic nature of our approach.

| # Transformer Blocks | # Attention Heads | Hidden Size | # Epochs |
|:---:|:---:|:---:|:---:|
| 8 | 12 | 256 | 200 |

Table 1: Architectural details of out implementation of ViT (Dosovitskiy et al., 2020) for the CIFAR10 dataset. Note that the training process (except the number of epochs, which is listed above) follows the Sect. A.3.

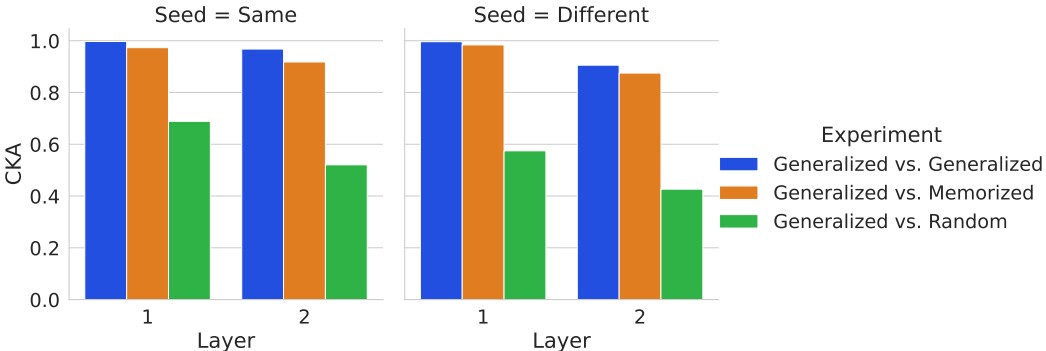

Figure 15: A layer wise comparison between a generalized, memorized, and randomly populated network using either (left figure) the same random seed or (right figure) different random seeds. This comparison reveals that, in either case (with same or different random seeds) early layers of these networks achieve relatively high CKA values.

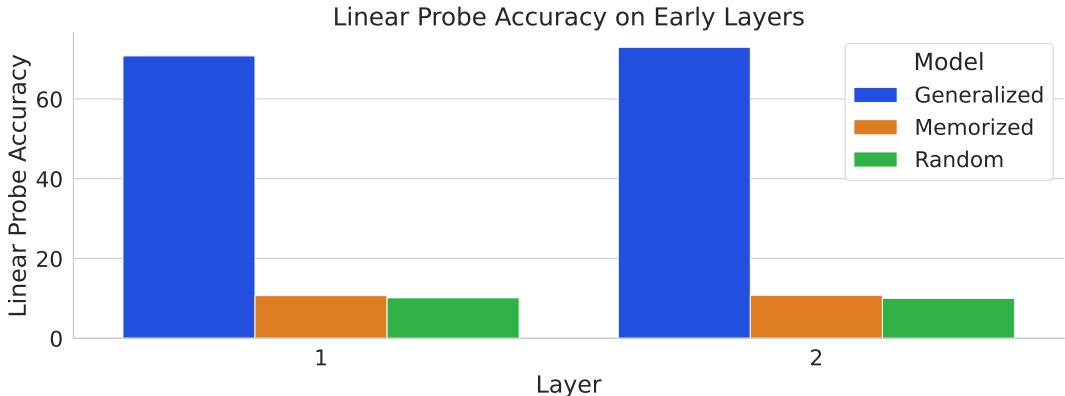

Figure 16: The linear probe accuracy obtained on the CIFAR10 test set for the generalized, memorized, and randomly populated network seen in Fig. 2 and 3. The results shown in this figure along with the ones shown in Fig. 15 suggests that high values of CKA similarity between two networks does not necessarily translate to similarly useful features.

### B.6 CLOSER LOOK AT THE EARLY LAYERS

In this section, we extend the study of the behaviour of CKA over the early layers presented in Fig. 2 and 3. In Fig. 15, we can see a layer wise comparison between a generalized, memorized, and randomly populated network using either (Fig. 15-left) the same random seed or (Fig. 15-right) different random seeds. This comparison reveals that, in either case (with same or different random seeds) early layers of these networks achieve relatively high CKA values.

However, as it was shown in Fig. 3, high values of CKA similarity between two networks does not necessarily translate to more useful, or similar, captured features. In order to quantify the usefulness of the features captured by each network in Fig. 2 and 3, we follow the same methodology as used in Self-supervised Learning (Chen et al., 2020) and in the analysis of intermediate representations (Zeiler & Fergus, 2014b). We evaluate the adequacy of representations by an optimal linear classifier using training data from the original task, in this case the CIFAR10 training data. The test set accuracy obtained by the linear probe is used as a proxy to measure the usefulness of the representations. Fig. 16, shows the linear probe accuracy obtained on the CIFAR10 test set for the generalized, memorized, and randomly populated network seen in Fig. 2 and 3. The results shown in this figure along with the ones shown in Fig. 15 suggests that high values of CKA similarity between two networks does not necessarily translate to similarly useful features.

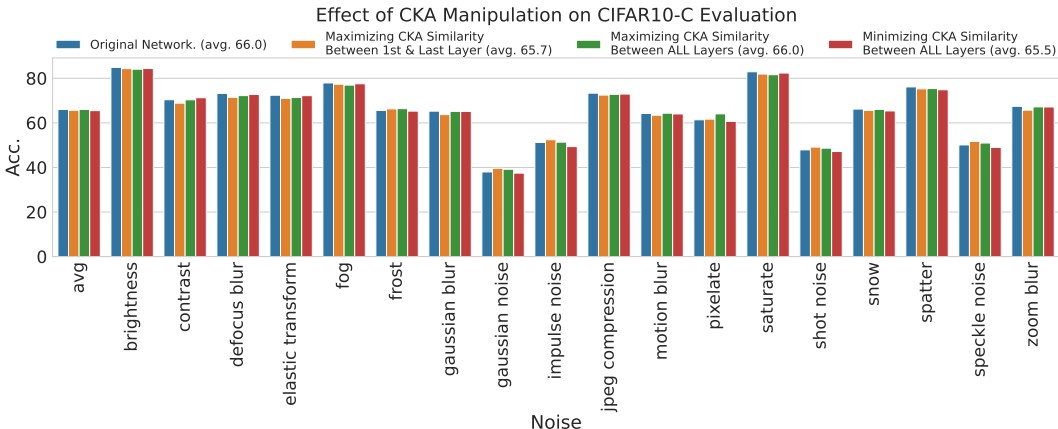

Figure 17: Results of OOD evaluation of models from Figure17 on CIFAR-10C (Hendrycks & Dietterich, 2018) corruptions data. We show results for each corruptions as well as the average results. We observe the performance remains largely unchanged.

## B.7 OUT OF DISTRIBUTION EVALUATION

In Sec. 4.3 we have showed that the CKA maps of models can be modified while maintaining an equivalent accuracy for the performance on the test set. We can also consider how the functionality of these models varies under other kinds of inputs. We thus consider evaluating the models created in Sec. 4.3, in particular those corresponding to Fig. 5 on the popular CIFAR-10 Corruptions datasets (Hendrycks & Dietterich, 2018). Figure 17 shows the results of this evaluation schema for the 3 different models explored in Fig. 5. We observe that overall the average performance of these models on this out-of-distribution data remains effectively unchanged. We emphasize that the models are trained only on uncorrupted data as in Sec. 4.3.

## C PROOFS

*Proof.* **Theorem 1**

First we introduce the notation $C_1 = S$ and $C_2 = X \backslash S$ and note that $C_1$ and $C_2$ form a partition of $X$, i.e. $X = C_1 \cup C_2$ with $C_1 \cap C_2 = \emptyset$. We note $C'_j$ the set of indices of $C_j$, meaning that $i \in C'_j \Leftrightarrow x_i \in C_j$. We then rewrite $X_{S,\vec{v},c}$ as being the union of the set of points in $C_1$ and the points in $C_2$ translated by $c$ in direction $\vec{v}$:

$$X_{S,\vec{v},c} = \{x : x \in C_1\} \cup \{x + c\vec{v} : x \in C_2\}$$

It is standard practice to center the two sets of representations being compared before using a representation similarity measures. $X$ is centered by hypothesis but $X_{S,\vec{v},c}$ is not. We first note that the mean of $X_{S,\vec{v},c}$ across all representations $\overline{X}_{S,\vec{v},c}$ is the vector:

$$
\begin{aligned}
\overline{X}_{S,\vec{v},c} &= \frac{1}{n} \sum_{x \in X_{S,\vec{v},c}} x \\
&= \frac{1}{n} \left( \sum_{i \in C'_1} x_i + \sum_{i \in C'_2} x_i + c\vec{v} \right) && \text{by definition of } X_{S,\vec{v},c} \\
&= \frac{1}{n} \left( \sum_{i \in C'_1 \cup C'_2} x_i + \sum_{i \in C'_2} c\vec{v} \right) \\
&= \frac{1}{n} \sum_{x \in X} x + \frac{1}{n} \sum_{i \in C'_2} c\vec{v} && \text{because } C_1, C_2 \text{ form a partition of } X \\
&= \frac{|C_2| c\vec{v}}{n} && \text{because } X \text{ is centered by hypothesis}
\end{aligned}
$$

From now on we note $Y$ as being the centered set of representations $X_{S,\vec{v},c}$ where we subtracted the mean $\overline{X}_{S,\vec{v},c}$ (here we used the fact that $|C_1| + |C_2| = n$):

$$Y = \left\{x - \frac{|C_2| c\vec{v}}{n} : x \in C_1\right\} \cup \left\{x + \frac{|C_1| c\vec{v}}{n} : x \in C_2\right\}$$

Now that we have workable expressions for $X$ and $X_{S,\vec{v},c}$ we focus on the computation of linear CKA which relies on the computation of three HSIC values: between $X$ and itself, between $Y$ and itself and between $X$ and $Y$:

$$\text{CKA}_{lin}(X,Y) = \frac{\text{HSIC}_{lin}(X,Y)}{\sqrt{\text{HSIC}_{lin}(X,X)\text{HSIC}_{lin}(Y,Y)}} \tag{5}$$

We also remind the reader that linear HSIC takes the form:

$$\text{HSIC}_{lin}(X,Y) = \frac{1}{(n-1)^2} tr(XX^\top YY^\top) = \frac{1}{(n-1)^2} \sum_{i=1}^{n}\sum_{j=1}^{n} \langle x_i, x_j \rangle \langle y_j, y_i \rangle \tag{6}$$

We can split the terms of the two sums into three distinct categories and compute the values of the inner products independently in terms of $x_i$, $x_j$ and $c$ for the three HSIC terms:

1. $i \in C_1'$ and $j \in C_1'$ (i.e. $x_i \in C_1$ and $x_j \in C_1$):

$$\langle x_i, x_j \rangle^2 = \langle x_i, x_j \rangle^2$$

$$\langle x_i, x_j \rangle \langle y_i, y_j \rangle = \langle x_i, x_j \rangle \langle x_i - \frac{c|C_2'|}{n} \vec{v}, x_j - \frac{c|C_2'|}{n} \vec{v} \rangle$$

$$= \langle x_i, x_j \rangle \left[ \langle x_i, x_j \rangle - \frac{c|C_2'|}{n} \langle x_i, \vec{v} \rangle - \frac{c|C_2'|}{n} \langle \vec{v}, x_j \rangle + \left( -\frac{c|C_2'|}{n} \right)^2 \langle \vec{v}, \vec{v} \rangle \right]$$

$$= \langle x_i, x_j \rangle \left[ \langle x_i, x_j \rangle - \frac{c|C_2'|}{n} \langle x_i, \vec{v} \rangle - \frac{c|C_2'|}{n} \langle \vec{v}, x_j \rangle + \frac{c^2|C_2'|^2}{n^2} \right]$$

$$= \langle x_i, x_j \rangle^2 - \frac{c|C_2'|}{n} \langle x_i, \vec{v} \rangle \langle x_i, x_j \rangle - \frac{c|C_2'|}{n} \langle \vec{v}, x_j \rangle \langle x_i, x_j \rangle + \frac{c^2|C_2'|^2}{n^2} \langle x_i, x_j \rangle$$

$$= \mathcal{O}(c) + \frac{c^2|C_2'|^2}{n^2} \langle x_i, x_j \rangle$$

$$\langle y_i, y_j \rangle^2 = \langle x_i - \frac{c|C_2'|}{n} \vec{v}, x_j - \frac{c|C_2'|}{n} \vec{v} \rangle^2$$

$$= \left[ \langle x_i, x_j \rangle - \frac{c|C_2'|}{n} \langle x_i, \vec{v} \rangle - \frac{c|C_2'|}{n} \langle \vec{v}, x_j \rangle + \frac{c^2|C_2'|^2}{n^2} \right]^2$$

$$= \mathcal{O}(c^3) + \frac{c^4|C_2'|^4}{n^4}$$

2. $i \in C_1'$ and $j \in C_2'$ (i.e. $x_i \in C_1$ and $x_j \in C_2$):

$$\langle x_i, x_j \rangle^2 = \langle x_i, x_j \rangle^2$$

$$\langle x_i, x_j \rangle \langle y_i, y_j \rangle = \langle x_i, x_j \rangle \langle x_i - \frac{c|C_2'|}{n} \vec{v}, x_j + \frac{c|C_1'|}{n} \vec{v} \rangle$$

$$= \langle x_i, x_j \rangle \left[ \langle x_i, x_j \rangle + \frac{c|C_1'|}{n} \langle x_i, \vec{v} \rangle - \frac{c|C_2'|}{n} \langle \vec{v}, x_j \rangle - \frac{c^2|C_1'||C_2'|}{n^2} \langle \vec{v}, \vec{v} \rangle \right]$$

$$= \langle x_i, x_j \rangle \left[ \langle x_i, x_j \rangle + \frac{c|C_1'|}{n} \langle x_i, \vec{v} \rangle - \frac{c|C_2'|}{n} \langle \vec{v}, x_j \rangle - \frac{c^2|C_1'||C_2'|}{n^2} \right]$$

$$= \langle x_i, x_j \rangle^2 + \frac{c|C_1'|}{n} \langle x_i, x_j \rangle \langle x_i, \vec{v} \rangle - \frac{c|C_2'|}{n} \langle x_i, x_j \rangle \langle \vec{v}, x_j \rangle - \frac{c^2|C_1'||C_2'|}{n^2} \langle x_i, x_j \rangle$$

$$= \mathcal{O}(c) - \frac{c^2|C_1'||C_2'|}{n^2} \langle x_i, x_j \rangle$$

$$\langle y_i, y_j \rangle^2 = \langle x_i - \frac{c|C_2'|}{n} \vec{v}, x_j + \frac{c|C_1'|}{n} \vec{v} \rangle^2$$

$$= \left[ \langle x_i, x_j \rangle + \frac{c|C_1'|}{n} \langle x_i, \vec{v} \rangle - \frac{c|C_2'|}{n} \langle \vec{v}, x_j \rangle - \frac{c^2|C_1'||C_2'|^2}{n^2} \right]^2$$

$$= \mathcal{O}(c^3) + \frac{c^4|C_1'|^2|C_2'|^2}{n^4}$$

3. $i \in C_2'$ and $j \in C_2'$ (i.e. $x_i \in C_2$ and $x_j \in C_2$):

$$\langle x_i, x_j \rangle^2 = \langle x_i, x_j \rangle^2$$

$$\langle x_i, x_j \rangle \langle y_i, y_j \rangle = \langle x_i, x_j \rangle \langle x_i + \frac{c|C_1'|}{n} \vec{v}, x_j + \frac{c|C_1'|}{n} \vec{v} \rangle$$

$$= \langle x_i, x_j \rangle \left[ \langle x_i, x_j \rangle + \frac{2c|C_1'|}{n} \langle x_i, \vec{v} \rangle + \left( + \frac{c|C_1'|}{n} \right)^2 \langle \vec{v}, \vec{v} \rangle \right]$$

$$= \langle x_i, x_j \rangle \left[ \langle x_i, x_j \rangle + \frac{2c|C_1'|}{n} \langle x_i, \vec{v} \rangle + \frac{c^2|C_1'|^2}{n^2} \right]$$

$$= \langle x_i, x_j \rangle^2 + \frac{2c|C_1'|}{n} \langle x_i, x_j \rangle \langle x_i, \vec{v} \rangle + \frac{c^2|C_1'|^2}{n^2} \langle x_i, x_j \rangle$$

$$= \mathcal{O}(c) + \frac{c^2|C_1'|^2}{n^2} \langle x_i, x_j \rangle$$

$$\langle y_i, y_j \rangle^2 = \langle x_i + \frac{c|C_1'|}{n} \vec{v}, x_j + \frac{c|C_1'|}{n} \vec{v} \rangle^2$$

$$= \left[ \langle x_i, x_j \rangle + \frac{2c|C_1'|}{n} \langle x_i, \vec{v} \rangle + \frac{c^2|C_1'|^2}{n^2} \right]^2$$

$$= \mathcal{O}(c^3) + \frac{c^4|C_1'|^4}{n^4}$$

When we take $\lim\limits_{c \to \infty} CKA_{lin}(X, Y) = \lim\limits_{c \to \infty} \frac{\text{HSIC}_{lin}(X,Y)}{\sqrt{\text{HSIC}_{lin}(X,X)\text{HSIC}_{lin}(Y,Y)}}$, it is easy to see that the terms with the highest powers of $c$ will dominate the expression. At the numerator that is $c^2$ and at the denominator that is $c^4$ inside the square root. To convince oneself of this it suffices to divide by $c^2$ at the numerator and at the denominator, all terms except the higher power ones will then tend to 0 as $c$ tends to infinity, so at the limit we have:

$$\lim_{c \to \infty} CKA_{lin}(X, Y)$$

$$= \lim_{c \to \infty} \frac{\text{HSIC}_{lin}(X, Y)}{\sqrt{\text{HSIC}_{lin}(X, X)\text{HSIC}_{lin}(Y, Y)}}$$

$$= \lim_{c \to \infty} \frac{\sum_{i=1}^{n} \sum_{j=1}^{n} \langle x_i, x_j \rangle \langle y_j, y_i \rangle}{\sqrt{\left( \sum_{i=1}^{n} \sum_{j=1}^{n} \langle x_i, x_j \rangle^2 \right) \left( \sum_{i=1}^{n} \sum_{j=1}^{n} \langle y_i, y_j \rangle^2 \right)}}$$

$$= \lim_{c \to \infty} \frac{\frac{c^2 |C_2'|^2}{n^2} \sum_{i \in C_1'} \sum_{j \in C_1'} \langle x_i, x_j \rangle - \frac{2c^2 |C_1'||C_2'|}{n^2} \sum_{i \in C_1'} \sum_{j \in C_2'} \langle x_i, x_j \rangle + \frac{c^2 |C_1'|^2}{n^2} \sum_{i \in C_2'} \sum_{j \in C_2'} \langle x_i, x_j \rangle}{\sqrt{\sum_{i=1}^{n} \sum_{j=1}^{n} \langle x_i, x_j \rangle^2} \sqrt{\sum_{i \in C_1'} \sum_{j \in C_1'} \frac{c^4 |C_2'|^4}{n^4} + \sum_{i \in C_1'} \sum_{j \in C_2'} \frac{2c^4 |C_1'|^2 |C_2'|^2}{n^4} + \sum_{i \in C_2'} \sum_{j \in C_2'} \frac{c^4 |C_1'|^4}{n^4}}}$$

$$= \lim_{c \to \infty} \frac{\frac{c^2 |C_2'|^2}{n^2} \sum_{i \in C_1'} \sum_{j \in C_1'} \langle x_i, x_j \rangle - \frac{2c^2 |C_1'||C_2'|}{n^2} \sum_{i \in C_1'} \sum_{j \in C_2'} \langle x_i, x_j \rangle + \frac{c^2 |C_1'|^2}{n^2} \sum_{i \in C_2'} \sum_{j \in C_2'} \langle x_i, x_j \rangle}{\sqrt{\sum_{i=1}^{n} \sum_{j=1}^{n} \langle x_i, x_j \rangle^2} \sqrt{\frac{c^4 |C_2'|^4 |C_1'|^2}{n^4} + \frac{2c^4 |C_1'|^3 |C_2'|^3}{n^4} + \frac{c^4 |C_1'|^4 |C_2'|^2}{n^4}}}$$

$$= \lim_{c \to \infty} \frac{\frac{c^2 |C_2'|^2}{n^2} \sum_{i \in C_1'} \sum_{j \in C_1'} \langle x_i, x_j \rangle - \frac{2c^2 |C_1'||C_2'|}{n^2} \sum_{i \in C_1'} \sum_{j \in C_2'} \langle x_i, x_j \rangle + \frac{c^2 |C_1'|^2}{n^2} \sum_{i \in C_2'} \sum_{j \in C_2'} \langle x_i, x_j \rangle}{\sqrt{\sum_{i=1}^{n} \sum_{j=1}^{n} \langle x_i, x_j \rangle^2} \sqrt{\frac{c^4 |C_2'|^2 C_1'|^2}{n^4} \left( |C_2'|^2 + 2|C_1'||C_2'| + |C_1'|^2 \right)}}$$

$$= \lim_{c \to \infty} \frac{\frac{c^2 |C_2'|^2}{n^2} \sum_{i \in C_1'} \sum_{j \in C_1'} \langle x_i, x_j \rangle - \frac{2c^2 |C_1'||C_2'|}{n^2} \sum_{i \in C_1'} \sum_{j \in C_2'} \langle x_i, x_j \rangle + \frac{c^2 |C_1'|^2}{n^2} \sum_{i \in C_2'} \sum_{j \in C_2'} \langle x_i, x_j \rangle}{\frac{c^2 |C_2'||C_1'|}{n^2} \sqrt{\sum_{i=1}^{n} \sum_{j=1}^{n} \langle x_i, x_j \rangle^2} \sqrt{|C_2'|^2 + 2|C_1'||C_2'| + |C_1'|^2}}$$

$$= \lim_{c \to \infty} \frac{\frac{|C_2'|}{|C_1'|} \sum_{i \in C_1'} \sum_{j \in C_1'} \langle x_i, x_j \rangle - 2 \sum_{i \in C_1'} \sum_{j \in C_2'} \langle x_i, x_j \rangle + \frac{|C_1'|}{|C_2'|} \sum_{i \in C_2'} \sum_{j \in C_2'} \langle x_i, x_j \rangle}{\sqrt{\sum_{i=1}^{n} \sum_{j=1}^{n} \langle x_i, x_j \rangle^2} \sqrt{|C_2'|^2 + 2|C_1'||C_2'| + |C_1'|^2}}$$

$$= \frac{\frac{|C_2'|}{|C_1'|} \sum_{i \in C_1'} \sum_{j \in C_1'} \langle x_i, x_j \rangle - 2 \sum_{i \in C_1'} \sum_{j \in C_2'} \langle x_i, x_j \rangle + \frac{|C_1'|}{|C_2'|} \sum_{i \in C_2'} \sum_{j \in C_2'} \langle x_i, x_j \rangle}{\sqrt{\sum_{i=1}^{n} \sum_{j=1}^{n} \langle x_i, x_j \rangle^2} \sqrt{|C_2'|^2 + 2|C_1'||C_2'| + |C_1'|^2}}$$

$$= \frac{\frac{|C_2'|}{|C_1'|} \sum_{i \in C_1'} \sum_{j \in C_1'} \langle x_i, x_j \rangle - 2 \sum_{i \in C_1'} \sum_{j \in C_2'} \langle x_i, x_j \rangle + \frac{|C_1'|}{|C_2'|} \sum_{i \in C_2'} \sum_{j \in C_2'} \langle x_i, x_j \rangle}{\sqrt{\sum_{i=1}^{n} \sum_{j=1}^{n} \langle x_i, x_j \rangle^2} \sqrt{\left( |C_2'| + |C_1'| \right)^2}}$$

$$= \frac{\frac{|C_2'|}{|C_1'|} \sum_{i \in C_1'} \sum_{j \in C_1'} \langle x_i, x_j \rangle - 2 \sum_{i \in C_1'} \sum_{j \in C_2'} \langle x_i, x_j \rangle + \frac{|C_1'|}{|C_2'|} \sum_{i \in C_2'} \sum_{j \in C_2'} \langle x_i, x_j \rangle}{\sqrt{\sum_{i=1}^{n} \sum_{j=1}^{n} \langle x_i, x_j \rangle^2} \sqrt{n^2}}$$

$$= \frac{\frac{|C_2'|}{|C_1'|} \sum_{i \in C_1'} \sum_{j \in C_1'} \langle x_i, x_j \rangle - 2 \sum_{i \in C_1'} \sum_{j \in C_2'} \langle x_i, x_j \rangle + \frac{|C_1'|}{|C_2'|} \sum_{i \in C_2'} \sum_{j \in C_2'} \langle x_i, x_j \rangle}{n \sqrt{\sum_{i=1}^{n} \sum_{j=1}^{n} \langle x_i, x_j \rangle^2}}$$

If we look directly at the numerator, by linearity of the inner product we have:

$$\frac{|C_2'|}{|C_1'|} \sum_{i \in C_1'} \sum_{j \in C_1'} \langle x_i, x_j \rangle - 2 \sum_{i \in C_1'} \sum_{j \in C_2'} \langle x_i, x_j \rangle + \frac{|C_1'|}{|C_2'|} \sum_{i \in C_2'} \sum_{j \in C_2'} \langle x_i, x_j \rangle$$

$$= |C_1'||C_2'| \langle \frac{1}{|C_1'|} \sum_{i \in C_1'} x_i, \frac{1}{|C_1'|} \sum_{j \in C_1'} x_j \rangle - 2|C_1'||C_2'| \langle \frac{1}{|C_1'|} \sum_{i \in C_1'} x_i, \frac{1}{|C_2'|} \sum_{i \in C_2'} x_j \rangle$$

$$+ |C_1'||C_2'| \langle \frac{1}{|C_2'|} \sum_{i \in C_2'} x_i, \frac{1}{|C_2'|} \sum_{j \in C_2'} x_j \rangle$$

$$= |C_2'||C_1'| \left[ \langle \overline{x_1}, \overline{x_1} \rangle - 2 \langle \overline{x_1}, \overline{x_2} \rangle + \langle \overline{x_2}, \overline{x_2} \rangle \right]$$

$$= |C_2'||C_1'| \| \overline{x_1} - \overline{x_2} \|^2$$

Where $\overline{x_j} = \mathbb{E}_{x \in C_j}[x] = \frac{1}{|C_j'|} \sum_{i \in C_j'} x_i$ is the mean of the points in $C_j$. At the denominator we can multiply by $\frac{n}{n}$ to obtain:

$$n \frac{n}{n} \sqrt{\sum_{x_i \in X} \sum_{x_j \in X} \langle x_i, x_j \rangle^2} = n^2 \sqrt{\sum_{x_i \in X} \sum_{x_j \in X} \frac{1}{n^2} \langle x_i, x_j \rangle^2}$$

$$= n^2 \sqrt{\sum_{i=1}^n \lambda_i^2}$$

We note that the term with the square root function is the Frobenius norm of the Gram matrix of the data (matrix of inner products) $XX^\top$ multiplied by $\frac{1}{n}$ which, in turn, is equal to the square root of the sum of it's squared eigenvalues, where $\lambda_i$ is the $i$-th eigenvalue of the matrix $\frac{1}{n} XX^\top$. However, through singular value decomposition, the Gram matrix (multiplied by $\frac{1}{n}$) has the same eigenvalues as the (biased) covariance matrix of the data, i.e. $\frac{1}{n} X^\top X$. Using the notation $X_{i,j}$ to go over the rows (representations) of $X$ with $i$ and over the columns (dimensions or neurons) of $X$ with $j$ we can write the variance in the data as the sum of the variances from all dimensions:

$$Var(X) = \sum_{j=1}^p Var(X_{:,j})$$

$$= \sum_{j=1}^p \sum_{i=1}^n \frac{1}{n} (X_{i,j} - \overline{X_{:,j}})^2$$

$$= \frac{1}{n} \sum_{j=1}^p \sum_{i=1}^n X_{i,j}^2 \qquad \text{because the data is centered}$$

$$= \frac{1}{n} \sum_{i=1}^n \sum_{j=1}^p X_{i,j}^2$$

$$= \mathbb{E}_i[\|X_{i,:}\|^2]$$

$$= \mathbb{E}_{x \in X}[\|x\|^2]$$

$$= \sum_{l=1}^p \lambda_i$$

We can then write the denominator as:

$$n^2\sqrt{\sum_{i=1}^{n}\lambda_i^2} = n^2\frac{Var(X)}{Var(X)}\sqrt{\sum_{i=1}^{n}\lambda_i^2}$$

$$= n^2 Var(X)\frac{\sqrt{\sum_{i=1}^{n}\lambda_i^2}}{Var(X)}$$

$$= n^2\mathbb{E}_{x\in X}[\|x\|^2]\frac{\sqrt{\sum_{i=1}^{n}\lambda_i^2}}{\sum_{i=1}^{n}\lambda_i}$$

$$= n^2\mathbb{E}_{x\in X}[\|x\|^2]PR(X)^{-1/2}$$

And we can rewrite the whole expression as:

$$\lim_{c\to\infty}CKA_{lin}(X,Y) = \frac{|C_1'||C_2'|\|\overline{x_1}-\overline{x_2}\|^2\sqrt{PR(X)}}{n^2\mathbb{E}_{x\in X}[\|x\|^2]}$$

Where $PR(X)$ is the participation ratio, an effective dimensionality estimator often used in the literature and is defined as:

$$PR(X) = \frac{\left(\sum_{i=1}^{p}\lambda_i\right)^2}{\sum_{i=1}^{p}\lambda_i^2}$$

With $\lambda_i$ being the $i$-th eigenvalue of the covariance matrix of the data $X$. We make the replacements $\frac{|C_1|}{n} = \frac{|S|}{n} = \rho$ and $\frac{|C_2|}{n} = \frac{n-|C_1|}{n} = 1-\rho$. Also, because the data is centered we have $|C_1|\overline{x_1} + |C_2|\overline{x_2} = 0$ and we can isolate $\overline{x_2} = \frac{-|C_1|\overline{x_1}}{|C_2|}$ so we have:

$$\|\overline{x_1}-\overline{x_2}\|^2 = \|\overline{x_1} + \frac{|C_1|\overline{x_1}}{|C_2|}\|^2$$

$$= (1 + \frac{|C_1|}{|C_2|})^2\|\overline{x_1}\|^2$$

$$= (1 + \frac{n|C_1|}{n|C_2|})^2\|\overline{x_1}\|^2$$

$$= (1 + \frac{\rho}{1-\rho})^2\|\overline{x_1}\|^2$$

We then define $\Gamma$ to contain all terms of $\rho$:

$$\Gamma(p) = \rho(1-\rho)(1 + \frac{\rho}{1-\rho})^2$$

$$= \frac{\rho}{1-\rho}$$

The following bounds hold: $\Gamma(\rho)\in(0,1]$ for $\rho\in(0,0.5]$ reached when $\rho = \frac{1}{2}$. Finally, we get the final expression by changing $\overline{x_1} = \mathbb{E}_{x\in C_1}[x] = \mathbb{E}_{x\in S}[x]$:

$$\lim_{c\to\infty}CKA_{lin}(X,Y) = \Gamma(\rho)\frac{\|\mathbb{E}_{x\in S}[x]\|^2}{\mathbb{E}_{x\in X}[\|x\|^2]}\sqrt{PR(X)}$$

$\square$

*Proof.* **Corollary 2**

To prove Corollary 2 it suffices to note that the fact that $\rho = \frac{|S|}{|X|}$ is in $(0, \frac{1}{2}]$ is not used anywhere in the proof of Thm. 1 other than to derive the bounds for $\Gamma(\rho)$. We can then conclude that Thm. 1 still holds if $S$ is taken such that $\rho = \frac{|S|}{|X|} \in (0.5, 1)$ only with different bounds for $\Gamma(\rho)$.

We note however that the expression in Thm. 1 can be written in terms of $S$ and $\rho$ or in terms of $S' = X \backslash S$ and $\rho' = \frac{|S'|}{|X|}$ interchangeably. We first note that $\rho' = 1 - \rho$ and for simplicity purposes we will use the notation $\overline{S} = \mathbb{E}_{x \in S}[x]$ and $\overline{S'} = \mathbb{E}_{x \in S'}[x]$. We recall that the expression in Thm. 1 is:

$$\lim_{c \to \infty} CKA_{lin}(X, X_{S, \vec{v}, c}) = \Gamma(\rho) \frac{\|\overline{S}\|^2}{\mathbb{E}_{x \in X}[\|x\|^2]} \sqrt{PR(X)}$$

The term $\frac{\sqrt{PR(X)}}{\mathbb{E}_{x \in X}[\|x\|^2]}$ does not depend on the choice of using $S$ or $X \backslash S$ so we can focus on the rest:

$$\Gamma(\rho') \|\overline{S'}\|^2 = \rho'(1 - \rho')(1 + \frac{\rho'}{1 - \rho'})^2 \|\overline{S'}\|^2$$

$\rho'(1 - \rho')$ is easily found to be equal to $(1 - \rho)\rho$ and for the rest we have:

$$
\begin{aligned}
(1 + \frac{\rho'}{1 - \rho'})^2 \|\overline{S'}\|^2 &= \|(1 + \frac{\rho'}{1 - \rho'})\overline{S'}\|^2 \\
&= \|\overline{S'} - \overline{S}\|^2 \\
&= \|\overline{S} - \overline{S'}\|^2 \\
&= \|\overline{S} + \frac{\rho}{1 - \rho}\overline{S}\|^2 \\
&= (1 + \frac{\rho}{1 - \rho})^2 \|\overline{S}\|^2
\end{aligned}
$$

Where we used the fact that the data is centered so $|S|\overline{S} + |S'|\overline{S'} = 0$ so we can isolate $\overline{S} = \frac{-|S'|\overline{S'}}{|S|}$. We conclude that we have:

$$\lim_{c \to \infty} CKA_{lin}(X, X_{S, \vec{v}, c}) = \Gamma(\rho') \frac{\|\overline{S'}\|^2}{\mathbb{E}_{x \in X}[\|x\|^2]} \sqrt{PR(X)}$$

$\square$

*Proof.* **Corollary 4**

We already have $x \in S \Rightarrow \langle w, x \rangle \leq k$. Pick any $\vec{v} \in \mathbb{R}^p$ that is orthogonal to $w$ and we have:

$$
\begin{aligned}
\langle w, x + c\vec{v} \rangle &= \langle w, x \rangle + c \langle w, \vec{v} \rangle && \text{by the linearity of the inner product} \\
&= \langle w, x \rangle && \text{since } \vec{v} \text{ is orthogonal to } w \\
&> k && \forall x \in X \backslash S
\end{aligned}
$$

$\square$

# D FURTHER DISCUSSION OF THEORETICAL RESULTS

## D.1 ANALYSIS OF THE TERMS IN THEOREM 1

In addition to the description of the terms that we provide in Section 3 (page 4 before the paragraph on CKA sensitivity to outliers) we go here into further detail regarding the terms from Thm. 1. The two expectation terms, $\mathbb{E}_{x \in X}[\|x\|^2]$ and $\|\mathbb{E}_{x \in S}[x]\|^2$ are relatively simple to describe. They respectively represent the average squared norms of all representations in $X$ and the squared norm of the mean of $S$, the subset of representations that are not being translated. As mentioned in the main text, since most neural networks are trained using weight decay, the network parameters, and hence the resulting representations as well as these two quantities are biased towards small values in practice. $\Gamma(\rho) = \frac{\rho}{1 - \rho}$ is the ratio of the points in $S$, that are not translated, with respect to the points

in $X\backslash S$ that are translated. Finally, the last term $\sqrt{\dim_{PR}(X)}$ is the square root of the participation ratio, which is used as an effective dimensionality estimate for internal representations. It is defined as $\dim_{PR}(X) \triangleq \frac{(\sum_i \lambda_i)^2}{\sum_i \lambda_i^2} \in [1, p]$ where $\{\lambda_i\}$ are the eigenvalues of the covariance matrix of $X$. These eigenvalues are perhaps best understood through the lens of principal component analysis which uses the eigenvectors of the covariance matrix as principal components and the eigenvalues for describing the variability of the data which is found along the direction of it's corresponding eigenvector. In other words, the eigenvalues of the covariance matrix encode the amount of variance in the data in an orthogonal basis of the ambient space of the data points. The participation ratio is the ratio between the squared sum of eigenvalues and the sum of the squared eigenvalues. We describe the two extreme cases for intuition. If all the variance of the data comes from a single direction then only one of the eigenvalues will be non zero (WLOG $\lambda_j$) and therefore both the numerator and the denominator will be equal to $\lambda_i^2$ so the PR will be 1, i.e. the estimated dimensionality will be 1 which makes sense given that all the variability comes from a single direction. On the other hand, if the variance is equally distributed across all directions, all the eigenvalues will have the same value $\lambda$ and the participation ratio will be $\frac{(\sum_i \lambda)^2}{\sum_i \lambda^2} = \frac{p^2 \lambda^2}{p\lambda^2} = p$. Therefore the effective dimensionality will be the full dimensionality of the ambient space. Since there is an equal amount of variance in all directions, all dimensions of the ambient space "participate" equally in describing the dataset $X$.

## D.2 Additional intuition on Theorem 1 and Corollary 2

A question that may arise related to Corollary 2 is: what happens when $S$ is such that $\rho >> \frac{1}{2}$ and the value of $\Gamma$ quickly grows beyond 1? It turns out that it is the fraction of expectations that will cancel out large $\Gamma$ values. As a side note, before we start explaining the previous sentence, we point out that since $X$ contains a finite number of data points $\Gamma$ will never actually tend to infinity even if it can take on large values. Now suppose $S$ is such that $\rho >> \frac{1}{2}$, i.e. $S$ contains a majority of the points in $X$, therefore $\Gamma$ will have a large value. Since the points in $X$ are centered and since $S$ contains a majority of the points from $X$, this implies one of two things. Either:

1. The mean of the points in $S$ will be almost zero ($\mathbb{E}_{x \in S}[x] \approx \mathbf{0}$), therefore the numerator will be almost zero since it is the squared norm of this mean. This can easily occur if $S$ is a very large part of $X$ (which is centered) and the points in $X\backslash S$ are not outliers with large norms so the mean of $S$ will be very close to the mean of $X$, which is 0 in this case. The denominator on the other hand ($\mathbb{E}_{x \in X}[\|x\|^2]$) can be large since we take the squared norm of the points in $X$, which are all positive values, and only afterwards compute the expected value.

2. The small number of points in $X\backslash S$ will have very large norms relative to those in $S$ and will be positioned on the opposite side of the origin with respect to the mean of $S$ (so the mean of $X$ can be at the origin). The bigger $S$ is inside $X$, the bigger the norms of the points in $X\backslash S$ will need to be to compensate. Therefore, the numerator $\|\mathbb{E}_{x \in S}[x]\|^2$ will again be very small with respect to the denominator $\mathbb{E}_{x \in X}[\|x\|^2]$. At the denominator, since we take the squared norm of the points in $X$ before computing the expected value, the points in $X\backslash S$ which have very large norms compared to those in $S$ will dominate the expected value. Consequently, the value of the fraction of these two terms will be very small.

Another insightful way of analyzing this situation, which is perhaps more intuitive, is to look at it through the lens of the proof of Corollary 2. Given the fact that the data is centered before applying CKA, translating the points in $X\backslash S$ in any random direction or translating the points in $S$ in the exact opposite direction will yield the same set of representations post-centering, and thus will give the same CKA value. Therefore, for any $S$ such that $\rho > \frac{1}{2}$ we can look at the complementary case where we swap the definitions of $S$ and $X\backslash S$ in Theorem 1 and the translation direction is inverted (opposite direction) and we find ourselves with the new $\rho$ being smaller than $\frac{1}{2}$, the new $\Gamma$ smaller than 1 and Theorem 1 still applies. More explicitly we can apply the Theorem 1 to $S' = X\backslash S$ with $\rho' = (1 - \rho) < \frac{1}{2}$ and the new direction $\overrightarrow{v'} = -\overrightarrow{v}$.

