# OpenReview forum: "Reliability of CKA as a Similarity Measure in Deep Learning"
_ICLR.cc/2023/Conference — ICLR 2023 poster_

### Official Review · Reviewer_wkkL · 2022-10-19

**Confidence:** 4
**Correctness:** 3
**Technical Novelty And Significance:** 3
**Empirical Novelty And Significance:** Not applicable
**Recommendation:** 6

**Clarity, Quality, Novelty And Reproducibility:**

- This paper is nicely written and easy to follow, and contains the necessary background information of different neural representation similarity measures to be self-contained.
- There are conclusions that are not well-supported, including the natural occurrence of subset translation in normal neural networks training and inconsistent ways to judge the similarity of representations (visually or functionally).
- (Kornblith et al., 2019) showed that the CKA similarity is invariant to orthogonal transformations which preserve scalar products and Euclidean distances between examples. This implies that CKA is sensitive to changes in the Euclidean distances between examples. So the analysis in Section 3 is not quite surprising since the subset translation changes the Euclidean distances. That being said, the manipulation of the CKA map into any target map without impacting the model performance appears novel to me and is definitely an interesting contribution. A natural question here is: is this manipulation only applicable to CKA, or any existing similarity measure is susceptible as well?

**Strength And Weaknesses:**

On the strengths,
- The authors provide a formal characterization of the CKA sensitivity to the family of subset translation transformations (i.e., the translation of a subset of the representations) and demonstrate special cases of this family can preserve "functional behavior" of the learned neural representations. Thus, an ideal similarity measure should be insensitive to this family, but the CKA fails to do so.
- CKA is a family of measures so merely showing the linear case is sensitive is not very conclusive. It is nice that the authors not only conduct analysis on linear CKA but also nonlinear CKA with RBF kernels and show that the nonlinear ones still suffer from the same sensitivity.
- The demonstration of the CKA map can be manipulated into any target map without changing the model performance is quite interesting and reminds me of the adversarial examples. This result questions the validity of many existing conclusions regarding the similarity between different models and their behaviors.

On the weaknesses,
- My biggest concern is about the "similarity" advocated in this work. Informally, the similarity assumption is that models that get similar accuracies have similar representations. In order to make this more robust, the authors add more constraints including same linear separating hyperplane and same margin. Even with these additional constraints, I do not think this assumption is a good one. Models learned with different random seeds have almost identical in-distribution accuracies, but vastly different out-of-distribution accuracies. In an analogy, this suggests that two distributions having the same first-order moment should be the same distribution. But we only consider two distributions same if they share the same moments in all orders. Therefore, unless the fixed models have similar accuracies in *all* datasets might we decide them to have similar representations. Indeed, similar assumptions have been considered in the literature, but they are used in a different way. For example, (Ding et al., 2021) uses this as an audit. That is, representations having different in-domain and/or out-of-domain accuracies should be different. This is very different from what used in this work.
- Some of the conclusions are not well supported. For example, it is argued throughout the paper that the subset translation can *naturally* happen in training neural networks but fails to provide any concrete example. In Section 4.1, the conclusion of "considerably high CKA similarity values for early layers, does not necessarily translate to more useful, or similar, captured features" is based on the visual differences of the learned filters without considering the "functional behaviors" advocated in this work. It is possible that they are *functionally* similar even though they are visually different.

**Summary Of The Paper:**

This work cautions the application of CKA similarity measure in comparing neural representations in practice and points out a few issues including 1) its sensitivity to the family of the subset translations where certain transformations do not change the functional behaviors of the model; 2) its value can be directly manipulated without significant changes in the model behaviors.

**Summary Of The Review:**

The CKA similarity has been widely used to draw conclusions in comparing neural representations across model architectures and learning paradigms, thus it is critical to better understand its strengths and weaknesses. I agree with the authors that any similarity measure should not be blindly applied, and multiple measures should be considered to cover multiple notions of similarity. Even though the CKA map manipulations is quite interesting, I am concerned with the similarity assumption advocated in this work: models with similar accuracies (in one dataset) should have similar representations. Since the main part of this work builds on this assumption, I am leaner towards the negative side.

---

> ### Author Response · Authors · 2022-11-16
> **Response to Reviewer wkkL (2/2)**
>
> - *Regarding Section 4.1*: We thank the reviewer for pointing out that we have forgotten to mention an important experiment that is placed in the appendix (due to space constraints). Specifically, we go further than a simple visual comparison of the weight matrices as we agree with the reviewer that it does not constitute an in-depth analysis. **In Figure 16 of the appendix we make this comparison stronger by using linear probes to quantify the usefulness of the learned weight matrices, and observe that the generalized models learn significantly more useful weights as quantified by larger probing accuracies.** We have now mentioned these results in the main text.
>
> We hope we have adequately addressed all the reviewers' concerns, we think the modifications made improve our submissions so we thank the reviewer for the comments. We would be glad to discuss any points that have remained unclear.

---

> > ### Comment · Reviewer_wkkL · 2022-11-17
> > **Thanks for the response**
> >
> > Thanks for pointing to the additional experiment in the appendix regarding the comparisons in Section 4.1.

---

> ### Author Response · Authors · 2022-11-16
> **Response to Reviewer wkkL (1/2)**
>
> We thank the reviewer for the thoughtful comments and suggestions and for the kind words regarding readability and the strength of our results. We will address all points below:
>
> - *Regarding Basis on Accuracy*: We agree with the reviewer that in-distribution accuracy is only one criteria of interest regarding a model. However it is a major criteria in the deep learning community which drives the use of CKA analysis. **Indeed models of interest are typically only analyzed by the deep learning community for their internal behavior (whether with CKA or other techniques such as feature visualization (Zeiler et al. 2014), linear probes, transfer learning, etc.) if they achieve high accuracy.** Consider the set of works using CKA mentioned in the 3rd to last line on Page 1, where models are directly compared using the CKA (Vision transformers vs CNN in one case and Wide CNN vs Thin CNN in another). The primary interest in performing such comparisons is due to the accuracy gaps (or lack thereof) displayed by the trained versions of these models. Consider (Raghu et al. 2021), where the goal is to understand if ViT uses fundamentally different mechanisms to classify images compared to CNNs. The analysis widely uses CKA based comparisons as an off-the-shelf method to determine differences between these models (for example that lower layers are more similar to upper layers in ViT).  In the context of that paper (i.e. the model has a high performance with a more general architecture than CNN thus it is of interest to know how the two differ) replacing the trained vision transformer with one having equivalent accuracy should not change the analysis. **Yet our work suggests the analysis based on CKA, and therefore any resulting conclusions, can be easily changed with slight modifications to the parameters and no change to the in-distribution accuracy.** We agree with the reviewer that OOD accuracy is increasingly analyzed by the community, thus we performed additional experiments in Appendix B.7 on our models that were created in Sec 4.3, **finding that their out-of-distribution accuracy, similar to the in-distribution one, can be kept the same for wildly different CKA maps**. Suggesting that, for the relevant inputs, the model output and functionality have not changed despite changing the CKA results and their subsequent interpretations.
>
> - Furthermore, we draw the reviewers attention to Figure 10 of the appendix for additional results that highlight that our CKA map manipulations affect equivalently representations of OOD data. **Even on a drastically different dataset (Patch Camelyon) from the one the networks were trained on and the CKA maps were optimized on (CIFAR10), our results hold and the CKA map manipulations seem to be stable across domains.** We have made sure to better mention these results in the main text.
>
> - We thank the reviewer for mentioning (Ding et al. 2021), which does an excellent job of linking functional characteristics to representations “similarity” as measured by different measures. **We view our work as being complementary to theirs and while they consider a wider range of similarity measures and tasks, we go more in depth into CKA by providing theoretical results and extensive empirical evidence. Our choice is justified and necessary given the predominance of CKA as a (dis)similarity measure in deep learning.** We note that much of our analysis and tools (particularly from Sec 4.3) is transferable to other functional characteristics and similarity metrics, which is however beyond the scope of the present work.
>
> - *Regarding concrete examples of subset translation*: **In the paragraph “CKA sensitivity to outliers” we discuss several results in which subset translation, specifically for small subsets or the “outlier scenario”, occur naturally during training.** All papers discussing the block structure in the representations of wide networks are examples of this since (Nguyen et al. 2022) have found that the block structure is actually caused by a few data points that “dominate” the first principal components of the representations. In other words, these points are far from the other representations which constitutes a translation when comparing with other networks that do not have this outlier structure and where all representations are more grouped together. Similar observations are made in (Ding et al. 2021) who find that CKA is insensitive to changes in all but the top few PCs, in other words it is strongly affected by directions of high variance and therefore by any possible outliers.

---

> > ### Comment · Reviewer_wkkL · 2022-11-17
> > **The similarity assumption**
> >
> > I want to first thank the authors for their detailed response, and also point out that I myself did enjoy reading this paper and find the CKA map manipulation quite interesting and novel. The additional analysis on OOD data and transferability to a different dataset is reassuring as well.
> >
> > But unfortunately, I do not think the authors have fully addressed my major concern regarding their assumption of what a "sound" similarity measure should be. The story of this work is that **the CKA similarity measure is sensitive to the family of subset translation which in certain cases can preserve the model performance on a particular dataset, therefore we should be cautious using the CKA measure**. This assumption implies that **two presentations should be considered similar if they produce same performance on one dataset**. I do not think this assumption appropriate. There are a lot of examples of vastly different "**transfer learning**" performance of two presentations which produce comparable results in the in-domain data. Therefore, this would largely impact the merit of the work if the assumption itself is questionable.
> >
> > On the other hand, (Kornblith et al., 2019) showed that the CKA similarity is invariant to orthogonal transformations which preserve scalar products and Euclidean distances between examples. This implies that **CKA is sensitive to changes in the Euclidean distances between examples**. Therefore, the sensitivity to subset translation is not surprising since it changes the Euclidean distances.

---

> > > ### Author Response · Authors · 2022-11-18
> > > **Clarifying our assumptions**
> > >
> > > We thank the reviewer for their additional comments and for giving us an opportunity to clear out any misunderstandings.
> > >
> > > To directly address the reviewer's main concern, **we do not restrict ourselves to the context of a single dataset and the "same in-distribution accuracy means similarity” context is not an explicit assumption of our results**. Specifically, **our theoretical results are completely agnostic to the context one uses CKA in and what model functionality one considers, which constitutes one of their main advantages**. They hold in the standard "in-distribution" context, in OOD contexts, transfer learning contexts and any other context relevant to the ML community. These theoretical results might even prove useful in contexts that are not yet explored by ML researchers as of today.
> > >
> > > As the reviewer pointed out, there are examples of  different "transfer learning" performances of two representations (coming from two distinct models) which produce comparable results on the "in-domain" data. **Our results can help explain such situations.** For example, it is possible that on the "in-domain" data the two models achieve high accuracy and have very similar representations (high CKA). But perhaps one of the two models learns significantly less transferable features (weights or parameters) leading to a lot more outliers in the representation space, less similar representations (lower CKA) and worse performance on the "out-of-domain" or "transfer" data when compared to the other model. This specific scenario would be entirely explainable by our theoretical and empirical results, and it is only one among many other possible examples. Testing all possible examples and scenarios is outside the scope of our work (and infeasible in the general case) and we believe our theoretical results and empirical framework are presented in a general enough way as to be easily applicable to any other context in future work.
> > >
> > > Many  of our experiments are run on a single dataset with in-distribution accuracy as a focus only because it **constitutes the basic framework in which CKA is currently applied the  most frequently in the DL community**. Indeed, in the DL community, CKA similarity is often used to explain how certain models achieve high accuracy by comparing representations and inferring information by high similarity or low similarity. Our results show that high or low similarity in many of these cases may be independent of performance, or even output (logits, scores, our output class) of the model. **This is not a matter of whether a similarity measure itself is sound, but rather whether the deductive process of drawing conclusions from it is sound in the context in which CKA is being used.** A concrete example of this is provided in the rebuttal with the ViT vs. CNN paper in which the choice of analysis is driven entirely by accuracy associated with two model architectures achieving similar accuracy. However, as we previously pointed out we do also have OOD results on other datasets that support our conclusions. Also, our choice of analyzing this specific scenario due to its importance does not remove from the universality of our framework. We believe the "soundness" of a similarity measure depends on the context, and our results (and analysis techniques from Sec 4.3) are general enough to be applied to a broad spectrum of different contexts even if they can't all be explored here.
> > >
> > > Regarding the invariance to orthogonal transformations, we point out that simply stating that CKA is sensitive to changes in the Euclidean distances between examples and actually analytically characterizing the limiting behavior of CKA when subset translations occur do not provide the same amount of information. The latter is significantly more useful since it can provide insights into CKA's failure cases and can help pinpoint exactly what terms and what statistics are at fault. Furthermore, even if only empirical, our results show that even relatively small translation distances can significantly affect the CKA value, which is significantly more useful and specific than simply noting that CKA is sensitive to changes in the distances between examples. We also refer the reviewer to our response to concern 2 for Reviewer YJ8i which is somewhat related.
> > >
> > > We hope this clarifies certain aspects of our work and would be glad to go into further details if necessary.

---

> > > > ### Comment · Reviewer_wkkL · 2022-11-27
> > > > **Raised my score to 6**
> > > >
> > > > Thanks for the further clarification. As I mentioned earlier, I did like this work. The part that I have concern with is the story-telling/presentation of the "unreliability" of the CKA. That being said, I do find the clarification gives a much better presentation and would encourage the authors to consider incorporating this rationale into their revisions if possible.

---

> > > > > ### Author Response · Authors · 2022-12-01
> > > > > **Response to Reviewer wkkL**
> > > > >
> > > > > We are grateful for the reviewers careful consideration of our responses and for their willingness to engage in a discussion, as well as for raising their score after said discussion.
> > > > >
> > > > > Regarding the clarifications, we are glad the reviewer's found them useful. Fortunately, while it might not be possible to significantly change the draft at this point all the clarifications posted on OpenReview will be available to whomever is looking for additional insight into our work.

---

### Official Review · Reviewer_UWFM · 2022-10-25

**Confidence:** 4
**Correctness:** 3
**Technical Novelty And Significance:** 2
**Empirical Novelty And Significance:** 3
**Recommendation:** 6

**Clarity, Quality, Novelty And Reproducibility:**

I think the results are mostly novel, although they build off of previous works. One of the main conclusions, that CKA is sensitive seems to have been mentioned in passing by Nguyen et al. (2022), as the authors acknowledge. Further, the main take home message of Maheswaranathan et al. (2019) is that representational geometry is not one-to-one mapped to algorithms or functions computed by a network. However, the experiments (and to a some extent the theorems) in this paper make these observations more precise.

I also have to admit that I'm somewhat lukewarm about a few observations in this paper. For example, the result in Figure 4b seems pretty obvious to me, but the trick of translating points like this without effecting network function only works for the last layer in the network and so this doesn't end up being very insightful.

The theory portion of the paper is could be improved for clarity (see above).

The experimental portion of the paper is reasonably clear, though could also be improved. I recommend changing the colormaps on the bottom row of Figure 6 to match the colormap of the original network CKA map shown on the rightmost panel. This would help compare the bottom row across the different CKA targets.

**Strength And Weaknesses:**

I liked the motivation of the paper. CKA is an influential tool in the field and it feels important for us to closely examine its failure cases. I am unaware of any paper within the ML literature which takes a close look into these issues. Figure 2 and Figure 8 show interesting empirical results which I feel are highlights of this work.

On the downside. I did not feel like the main theorem proved in this paper gave me much intuition, and in fact some of the statements made by the authors were quite confusing to me. For example, at the top of page 5, the authors say "$\Vert \mathbb{E}[x] \Vert^2  / \mathbb{E}[ \Vert x\Vert^2]) \sqrt{\text{dim}_\text{PR}(X)}$ will be of relatively small value in practice so the whole expression in Eq.
2 will be dominated by $\Gamma(\rho)$." Maybe I'm missing something very basic but if $\Vert \mathbb{E}[x] \Vert^2  / \mathbb{E}[ \Vert x\Vert^2]) \sqrt{\text{dim}_\text{PR}(X)} \approx 0$ then you'd have $\text{CKA} \approx 0$ always in the limit of $c \rightarrow \infty$? Even when $\rho = 1/2$? More generally, I think that the limit of very large $c$ isn't too intuitive or sensible to think about (e.g. if some form of weight decay or other regularization is applied). It would be more useful to understand the rate at which the CKA decreases (for finite $c$) in Figure 4. Perhaps the authors can clear up their intuitions in the rebuttal period.

I also think that the results in this paper could be made more rigorous. For example, Figure 2 intrigues me, but I would love to see how much CKA varies across different random seeds. In particular, I would like to see additional lines for `Generalized vs Generalized` and `Memorized vs Memorized` and `Randomized vs Randomized` across random seeds. Also, error bars indicating standard deviations.

**Summary Of The Paper:**

The authors inspect the robustness of centered kernel alignment (CKA) as a measure of representational similarity. They analytically derived the limiting behavior of CKA as a subset of points are translated infinitely far away. They suggest that CKA is sensitive to a small number of "outlier" values. They empirically back up their theory by showing it is possible for most networks to achieve a variety of across-layer CKA maps without hurting performance. Further, CKA similarity between networks memorizing and generalizing on CIFAR-10 images was surprisingly high. Thus, their empirically results show that representational similarity as quantified by CKA can be decoupled from network function.

**Summary Of The Review:**

Overall, I find myself on the fence. I think in its current form the paper doesn't yet meet the bar for acceptance, but I am open to raising my score if the authors provide further empirical detail on Figure 2A (see comments above), and if they are able to revise their theory section to impart more intuitive insight.

---

> ### Author Response · Authors · 2022-11-16
> **Response to Reviewer UWFM**
>
> We thank the reviewer for acknowledging the importance of the subject of our work and for the insightful comments. We address all the reviewer’s concerns below and have made the necessary modifications to improve the text.
> - The reviewer’s intuition is correct, if the fraction of expectations multiplied by the PR dimensionality term is approximately 0 then CKA will be approximately 0 even if $\rho = \frac{1}{2}$. This is in fact another way in which the CKA value can be brought arbitrarily low by increasing the translation distance $c$. In other words, **it might not even be necessary for $\Gamma$ to be small if the rest of the terms are already close to zero**. We hope that this clarifies the theoretical results and we note that this only further supports our conclusions. Due to space constraints we cannot go into more details regarding the theoretical results in the main text. **However we have added a new section in the appendix where we describe in more detail the terms that appear in Thm. 1, their meaning and possible interactions and we provide additional interpretations of our theoretical results.**
> - *Regarding the rate of convergence*: We agree the rate of convergence is beneficial, yet results at the limit of very large c can also prove very useful for explaining empirical results and give intuition to CKA’s behavior. **We emphasize that in our work we see empirically that the rate of convergence is relatively fast** since in both Figures 4 (a) and (b, blue line) the CKA value drops to close to zero almost immediately for relatively small values of c (c < 100), and stays there. **So the limit value predicted by our theoretical result is reached quickly, i.e. for translation distances that are in the same order of magnitude as the norm of the representations considered**. While we agree that the results in Figure 4 aren’t necessarily surprising, the strength of our work in relation to these results is that we have theoretically predicted that this behavior will occur. We further note that the importance of our theoretical results is also emphasized by the results presented in Figure 8, which was pointed out by the reviewer as one of the highlights of this work. Specifically, **even without specifying exactly which transformations to implement to manipulate the CKA maps, the optimization process choses to translate a set of the representations to drop the CKA values**. This is perfectly inline with our theoretical results and further shows that the translation distance does not need to be infinitely large for the CKA values to be significantly altered.
> 	Regarding the suggestion that models trained with regularization may be immune to such effects, we note that CKA is typically measured on a test set, and outliers can still exist.
> - We thank the reviewer for the very constructive comments regarding Figure 2 and the associated experiments and feel the suggested modifications will improve the strength of our conclusions. **We have therefore added the results for multiple initializations (5 different random seeds), Figure 2 now contains the means and standard deviations over these multiple runs.**
>
> Again, we thank the reviewer for their constructive and insightful comments. We hope we have adequately addressed all the reviewers' worries and would be glad to further discuss any outstanding issues.

---

> > ### Comment · Reviewer_UWFM · 2022-11-16
> > **Interpreting the scale of $c$**
> >
> > I have a quick question -- you say "the CKA value drops to close to zero almost immediately for relatively small values of c (c < 100), and stays there."
> >
> > How do I interpret a value of $c = 100$? Could $c$ be normalized somehow (e.g. by the scale of $x$)?

---

> > > ### Author Response · Authors · 2022-11-17
> > > **Normalizing the translation distance by the average norm of the representations**
> > >
> > > We thank the reviewer for the great suggestion. We have normalized the translation distance $c$ by the average norm of the unmodified representations ($\mathbb{E}_{x\in X}[||x||]$) and have updated Figure 4, this makes the results presented easier to interpret. We see that both for the a and the b (blue line) subfigures, the CKA value drops significantly when the translation distance is of the same order of magnitude as the average norm of the representations. It seems to take larger values of c to drop the CKA in the outlier case, however there we only translate a single representation among multiple thousands. The results presented in Figure 8 would suggest that just moving a couple more representations, as opposed to only a single one, would be enough to drop the CKA value significantly (notice the small number of outliers in the top right of Fig. 8, right subfigure).

---

> > > > ### Comment · Reviewer_UWFM · 2022-11-25
> > > > **Raised my score to 6 (borderline accept)**
> > > >
> > > > After agonizing over this, I decided to bump my score up slightly. I think the practical experiments in this paper are interesting, and the authors did address some of my concerns in the initial review. Ultimately it is a close call and I can understand why the reviewer scores are high variance.
> > > >
> > > > I still think the theoretical portion of this paper gives me relatively little intuition and I'm not entirely convinced that the limit of $c \rightarrow \infty$ is approached very rapidly in practice. More theory on the rate of this convergence would be very useful. Also it would be interesting to theoretically understand how an adversarial agent with a fixed perturbation budget (e.g. constrained by a small number of points and/or total distance of movement) would optimally perturb representations to minimize or maximize CKA. These are suggestions for what kinds of theorems I would find more valuable and informative.

---

> > > > > ### Author Response · Authors · 2022-12-01
> > > > > **Response to Reviewer UWFM**
> > > > >
> > > > > We are grateful that the reviewer raised their score after our responses and we thank the reviewer for the insightful suggestions for future work.

---

### Official Review · Reviewer_U5e1 · 2022-10-25

**Confidence:** 4
**Clarity, Quality, Novelty And Reproducibility:** Please see other sections of the review.
**Correctness:** 4
**Technical Novelty And Significance:** 4
**Empirical Novelty And Significance:** 4
**Recommendation:** 8

**Strength And Weaknesses:**

Strengths

- The theoretical analysis on subset translations is insightful because it sheds light on why CKA is empirically sensitive to outliers ("dominant data points") and to transformations that preserve linear separability.

- The empirical results on explicitly optimizing CKA maps show that CKA is overly sensitive and may not be a useful metric to draw conclusions about general model behavior.

- The empirical findings cast doubt on the "block structure phenomenon" that is considered to distinguish representations of wide and narrow networks.


 Weaknesses:

- Specifics of Theorem 1 is a little hard to interpret.   A more intuitive description of these quantities after stating the theorem (e.g., eigenvalues of cov(X), rho, dim_PR etc) is missing.

- I think section 4.1 needs more work, or can be removed entirely. I am not sure how it relates to subset translation (the class of transformations studied in this paper) and the comparison is weak---comparing weight matrices visually does not make sense (even it "looks different")

**Summary Of The Paper:**

This paper focuses on understanding Centered Kernel Alignment (CKA), a common representational similarity metric. In particular, the paper shows that CKA is overly sensitive to a large class of "simple" representation-space transformations, which do not change the functional behavior of the trained models.  The theoretical analysis shows that CKA is sensitive to subset translation---a transformation that shifts a subset of data points in representation space. This finding sheds light on why CKA is empirically sensitive to outliers ("dominant data points") and to transformations that preserve linear separability.  The paper also shows that one can explicitly optimize the CKA map to match a comical target map without really changing the model behavior (e.g. accuracy) by much. They also show that the block structure phenomenon can be reproduced with this explicit optimization approach. Taken together, these results show that the CKA is unreliable and can easily manipulated without substantial changes to the functional behavior of the models

**Summary Of The Review:**

I vote to accept this paper. The paper is easy to read and well organized. The paper focuses on an interesting problem and shows that the CKA metric is not a good proxy for functional behavior of models. Most findings in the paper are original/novel and the practical implications of the theoretical analysis is insightful.

---

> ### Author Response · Authors · 2022-11-16
> **Response to Reviewer U5e1**
>
> We thank the reviewer for the constructive comments and suggestions and for generous comments regarding the paper’s organization and readability. We address the two weaknesses bellow:
> - We agree with the reviewer that more details regarding the terms in Thm. 1 would be useful. **We have added a section in the appendix where we describe in more detail the terms that appear in Thm. 1 and we give more intuition as to their meaning and possible interactions.**
> - Regarding section 4.1. We do agree that our main results are contained in Sec 4.2 and Sec 4.3. We think however that the points made in that section are important and we wish to keep the section in the manuscript. However the reviewer has made us realize something important, namely we forget to mention an important result related to this section that is placed in the appendix (due to space constraints). Specifically, we go further than a simple visual comparison of the weight matrices as we agree with the reviewer that it does not constitute a very strong analysis. **In Figure 16 of the appendix we make this comparison stronger by using linear probes to quantify the usefulness of the learned weight matrices, and observe that the generalized models learn significantly more useful weights.** We have now mentioned these results in the main text.

---

### Official Review · Reviewer_YJ8i · 2022-10-27

**Confidence:** 4
**Clarity, Quality, Novelty And Reproducibility:** See the previous section
**Correctness:** 4
**Technical Novelty And Significance:** 2
**Empirical Novelty And Significance:** 2
**Recommendation:** 6

**Strength And Weaknesses:**

The most interesting point about this paper is characterizing the sensitivity of CKA to outliers. Although sensitivity of CKA to large transformation to a few data points is not surprising to me, characterization of this sensitivity can be helpful for future work.

Major comments:

1. My main concern is that the submission evaluates reliability of CKA for comparing representations without a clear regard for the exact purpose for using this measure. A measure of (dis)similarity like CKA captures certain information about the representation and is invariant to other transformations. Whether an invariance is a strength or weakness for this measure depends on the context and the purpose for using the similarity measure. For example, Kornblith et al. use CKA for gaining insights about the training process of neural networks (see section 2.1 in their paper) while Ramasesh et al. use CKA to measure representation drift in the context of catastrophic forgetting. Kornblith et al. argue that a proper measure of similarity in their study needs to be invariant to rotation. In the context of Ramasesh et al., the similarity measure should not be invariant to rotation, since a rotated representation will have a different separating hyperplane and will result in forgetting if the classifier is not adjusted for this rotation. The submission would have been stronger if, before stating that a property of CKA is a weakness, it clearly and explicitly stated a context where CKA is used in previous work and discussed why this property of CKA is a weakness in this context.

2. The finding that representations with different separating hyperplanes may have a CKA close to 1 is well-known. Konblith et al. already show that CKA is invariant to orthogonal transformations (which clearly can change the separating hyperplane) and, as I said above, argue that this is a strength of CKA is their context.

3. Corollary 2 states that Theorem 1 holds for values of rho close to 1. In this case, Gamma will tend to infinity. Which factor in Theorem 1 will cancel out this large value of Gamma and keep CKA in [0,1]?

4. It looks like the definition of S has changed in Corollary 3. In theorem 1 it is the set of unmodified data points but corollary 3 it is the set of modified data points.

5. In the experiment for Figure 4a, the points from the second cube are translated in a random direction. The text claims that the cubes are still separated after this transformation. This is not true if the translation direction happens to be towards the first cube.

Minor comment: Please explain in the caption for Figure 3 how the plots are generated and what the colors represent.

**Summary Of The Paper:**

The submission discusses some pitfalls of using CKA as a similarity measure. In particular, the authors show that CKA is sensitive to outliers and some transformations that preserve linear separability and margin.

**Summary Of The Review:**

My main concern is that the submission mentions certain properties of CKA as weaknesses without discussing the context in which CKA is used and why these properties are weaknesses in these context. I also have some doubts about the correctness of some theoretical results and novelty of some empirical results that I elaborated on in the strengths and weaknesses section.

---

> ### Author Response · Authors · 2022-11-16
> **Response to Reviewer YJ8i (2/2)**
>
> Here we address the remaining comments:
> - *Reply Concern 2*: We agree with the reviewer that representations with different separating hyperplanes may have a CKA close to 1, this follows from CKA’s invariance to orthogonal transformations. **However the point we make in the paper is the opposite one. Namely, representations sharing the exact same separating hyperplanes, a highly relevant notion of similarity in the context of classification, can have a CKA value close to 0 even if they could be correctly classified by the exact same model.** This does not follow from the invariance to orthogonal transformations and is a novel point.
> - *Reply  on Concern 3*. **To directly answer the reviewer’s question regarding Corollary 2, it is the fraction of expectations that will cancel out large $\Gamma$ values.** As a side note, before going deeper into our answer, we point out that since $X$ contains a finite number of data points $\Gamma$ will never actually tend to infinity even if it can take on large values. Now suppose $S$ is such that $\rho >> \frac{1}{2}$, i.e. $S$ contains a majority of the points in $X$, therefore $\Gamma$ will have a large value. Since the points in $X$ are centered and since $S$ contains a majority of the points from $X$, this implies one of two things. Either:
>     1. The mean of the points in $S$ will be almost zero ( $\mathbb{E}\_{x \in S}[x] \approx \mathbf{0}$ ), therefore the numerator will be almost zero since it is the squared norm of this mean. This can easily occur if $S$ is a very large part of $X$ (which is centered) and the points in $X\backslash S$ are not outliers with large norms so the mean of $S$ will be very close to the mean of $X$, which is 0 in this case. The denominator on the other hand ($\mathbb{E}\_{x\in X}[||x||^2]$) can be large since we take the squared norm of the points in $X$, which are all positive values, and only afterwards compute the expected value.
>     2. The small number of points in $X\backslash S$ will have very large norms relative to those in $S$ and will be positioned on the opposite side of the origin with respect to the mean of $S$ (so the mean of $X$ can be at the origin). The bigger $S$ is inside $X$, the bigger the norms of the points in $X\backslash S$ will need to be to compensate. Therefore, the numerator $||\mathbb{E}\_{x\in S}[x] ||^2$ will again be very small with respect to the denominator $\mathbb{E}_{x\in X}[||x||^2]$. At the denominator, since we take the squared norm of the points in $X$ before computing the expected value, the points in $X\backslash S$ which have very large norms compared to those in $S$ will dominate the expected value. Consequently, the value of the fraction of these two terms will be very small.
> - **Another insightful way of analyzing this situation, which is perhaps more intuitive, is to look at it through the lens of the proof of Corollary 2.** Given the fact that the data is centered before applying CKA, translating the points in $X\backslash S$ in any random direction or translating the points in $S$ in the exact opposite direction will yield the same set of representations post-centering, and thus the same CKA value. Therefore, for any $S$ such that $\rho > \frac{1}{2}$ we can look at the complementary case where we swap the definitions of $S$ and $X\backslash S$ in Theorem 1 and the translation direction is inverted (opposite direction) and we find ourselves with the new $\rho$ being smaller than $\frac{1}{2}$, the new $\Gamma$ smaller than 1 and Theorem 1 still applies. More explicitly we can apply the Theorem 1 to $S' = X\backslash S$ with $\rho'=(1-\rho)<\frac{1}{2}$ and the new direction $\vec{v'}=-\vec{v}$.
> We have added this analysis to the appendix and mentioned it in the main text since it gives important insights into our main theoretical results.
> 4. Given the proof of Corollary 2 and the symmetric way in which Thm. 1 can be applied to either $S$ or $X\backslash S$ being translated, **the result does not change if the outlier is the point being translated in one direction or if all other points are translated in the opposite direction**. Centering will make both sets of representations exactly the same before the application of CKA. We chose to keep the current form of Corollary 3 for simplicity and clarity purposes.
> 5. That is correct, we verify that the random vector’s component in the axis along which the two cubes are separated is positive, i.e. the translated cube can only go away from the other one in that direction therefore remaining separate.
> 6. We have updated the caption of Figure 3 to further explain how it was generated.
>
> We really hope we have suitably addressed all of the reviewer’s concerns and we would gladly go into more details if there are any outstanding issues.

---

> > ### Comment · Reviewer_YJ8i · 2022-11-27
> > **Raising my score**
> >
> > Thanks for the detailed response. It addresses _all_ of the major comments and I will change my vote to accept. I agree with the other reviews that characterizing the behavior of CKA for infinitely large transformation is not very insightful but, given the careless use of CKA in the literature, I'd like to see papers raising awareness about the pitfalls.
> >
> > Regarding Fig 3: I'm not very familiar with this form of feature map but it seems like the differences in feature maps in this figure can be largely due to rotation. If this is the case, it is already known that these differences won't reduce CKA between the two representations and so this figure doesn't add new insights. (I'm just pointing this out for the authors to decide if they want to change or exclude this figure. This comment is not affecting my score.)

---

> > > ### Author Response · Authors · 2022-12-01
> > > **Response to Reviewer YJ8i**
> > >
> > > We thank the reviewer for having carefully considered our initial response. We are glad that our response adequately and clearly addressed all of the reviewer's major concerns and are grateful for the increase in the score.
> > >
> > > Finally we'll take into consideration the reviewer's comment regarding Fig. 3 for the (possible) final version as we can no longer change the draft at this point.

---

> ### Author Response · Authors · 2022-11-16
> **Response to Reviewer YJ8i (1/2)**
>
> We thank the reviewer for the insightful  and relevant comments. Regarding “Context”:
>
> We largely agree with the reviewer’s comment. **Whether an invariance or a sensitivity of a measure of (dis)similarity like CKA is a strength or a weakness does indeed depend on the context and the purpose for using the similarity measure.**  In fact, the importance of context  has been one of the main motivations for this work and is why we feel this work is so important to the deep learning scientific community right now.
>
> These measures are mathematical tools that aim to compare the shapes of two sets of representations. Each measure will define “shape” differently and will give more importance to some statistics of the representations or some geometrical and topological characteristics of the representations (sensitivity) and disregard others (invariances) while trying to boil down all this information into a single scalar. We completely agree with the reviewer that papers should justify their use of CKA or other (dis)similarity measures by relating the measure’s sensitivities and invariances to the specific statistical, geometrical or topological characteristics they wish to compare between representations. Many works applying CKA in the DL community mentioned in our introduction apply it without providing this context (see also reply to Reviewer wkkL), exactly motivating our paper. They use CKA as an “off the shelf” method to compare representations and then draw conclusions regarding their “similarity” without ever specifying what exact notion of similarity is considered. We do not aim to directly critique any particular work but provide a warning of such use of CKA.
>
> Our work is not aimed at saying that CKA is a “bad” measure of representation similarity and perhaps the word “weakness” which we used on two occasions in the manuscript was ill-chosen and should be replaced by “sensitivity”. This has been done in the revised text. Rather, **our work aims to characterize certain CKA sensitivities that are relevant in the *broad context* of deep learning and generalization** as justified throughout the text, especially in section 3. **We do not focus on one single context but rather discuss a multitude of contexts that can be relevant for deep learning research, such as linear separability by exact hyperplanes, margins of the decision boundaries, robustness to outliers, etc.**
>
> We wish to raise awareness to these sensitivities, specifically for the numerous cases where CKA is used without specifying the notions of similarity that are taken into consideration and why they are important in the context of deep learning. We acknowledge that in certain contexts, the exact notion of similarity that is relevant and linked to the deep learning concepts studied is unknown. **In such cases however, as we argue in the discussion, multiple (dis)similarity measures should be used and the results should be analyzed thoroughly and links with the sensitivities and invariances of the measures, such as those presented in our work, should be discussed.**

---

### Author Response · Authors · 2022-11-16
**General comment to all Reviewers addressing the scope of our work**

We thank the reviewers for all their constructive criticism and their insightful comments.

We wish to take this opportunity to discuss the motivation and scope of our work in the broader machine learning research context. As (dis)similarity measures such as CKA become more easily available and as more researchers become aware of their existence, these tools gain in popularity and are being used increasingly often to compare neural representations in numerous sub-fields of deep learning. **When focused on a specific problem, context or domain of application it is easy to forget that an individual (dis)similarity measure is not an end all be all tool to draw conclusions about the similarity of representations.** Each individual similarity measure focuses on different statistical and geometrical features of the representations and its “sensitivities” and “invariances” are directly related to the specific notion of “similarity” that is being quantified. Thorough theoretical and empirical analyses of these measures is necessary to understand these sensitivities and invariances. Furthermore, how specific statistical and geometrical features of the representations relate to the different model functionalities that a researcher might want to consider is still an open question.

**While all this may be clear to the reviewers, it is not necessarily obvious to the broader deep learning community.** Not all papers using CKA as a representation similarity measure discuss the method’s sensitivities or invariances and how the representation characteristics that are captured by it may be related to the model functionality. CKA is sometimes used as an off-the-shelf method to compare network representations and the notion of “similarity” that is being compared is not always explicitly addressed.

**Our work is not one in which CKA is used in a specific context or on a specific problem to study and draw conclusions about deep learning models. Instead, in our work CKA *is* the object of interest and we theoretically and empirically characterize some of its sensitivities and invariances that are important and relevant in the broad context of deep learning and generalization.** This does not mean, and we don’t say that CKA is a “bad” similarity measure or that there don’t exist contexts in which CKA’s sensitivities and invariances are desirable. Clearly such contexts do exist and there are papers that we cite that are examples of this. **However we strongly believe that a paper such as ours, which concretely addresses some of CKA’s sensitivities and discusses how they might not be desirable in some, albeit not all, contexts, is necessary. It offers the scientific community clear examples of cases where CKA does fail as a similarity measure. Finally, our work argues that one should be careful when using such measures and that a more methodological approach is necessary when comparing neural representations.**

---

### Decision · Program_Chairs · 2023-01-20

**Decision:**

Accept: poster

**Justification For Why Not Higher Score:**

The paper presents a theoretical analysis on the sensitivity of CKA to outliers and transformations. The paper does not warrant a higher score because:
1. Although the theoretical analysis are sound and correct, the original paper on CKT [1] had already implied that the transformations do not apply to all data sets.
2. The presentation of the paper needs to be improved for clarity.


Simon Kornblith, Mohammad Norouzi, Honglak Lee, Geoffrey Hinton, "Similarity of Neural Network Representations Revisited," ICML 2019.

**Justification For Why Not Lower Score:**

The theoretical analysis of the paper is interesting and the experiments are sound. The authors have addressed most of the comments raised by the reviewers.

**Metareview: Summary, Strengths And Weaknesses:**

The paper theoretical analyses the sensitivity of CKA to outliers and to transformations. Although the original paper on CKA has already implied that the transformations may not apply to all data sets and would preserve functional behavior in certain data sets, this paper shows this theoretically. The theory is good, but the authors should revise the paper and edit it for clarity. The authors have addressed the reviewers comments fairly and the experiments are sound

**Note From Pc:**

if the above contains the word "oral" or "spotlight" please see: "oral" presentation means -> notable-top-5% and "spotlight" means -> notable-top-25%. As stated in our emails, we are disassociating presentation type from AC recommendations

**Summary Of Ac-Reviewer Meeting:**

The following comments were shared by the reviewers at the AC-Reviewer meeting that was held on 29/11/2022:
1. The Theoretical parts are correct, but not very exciting. The original paper on CKT has already implied that the transformations may not apply to all data sets.
2. The paper should be presented with more clarity. The transformations would preserve functional behavior in certain data sets
3. The authors have addressed some of the concerns raised by the reviewers, although they need to clarify regarding the outputs. Authors have also agreed to reflect the rationality and assumptions in the revised manuscript
4. Although the experiments are sound, CKA could be manipulated to any kind of result you want.

The authors have responded to the re viewer comments after the meeting, and the reviewers have increased their scores after this.